# The Generative Leap: Tight Sample Complexity for Efficiently Learning Gaussian Multi-Index Models

**Alex Damian**
PACM
Princeton University

**Jason D Lee**
Electrical and Computer Engineering
Princeton University

**Joan Bruna**
Courant Institute and Center for Data Science
New York University

## Abstract

In this work we consider generic Gaussian Multi-index models, in which the labels only depend on the (Gaussian) $d$-dimensional inputs through their projection onto a low-dimensional $r = O_d(1)$ subspace, and we study efficient agnostic estimation procedures for this hidden subspace. We introduce the *generative leap* exponent, a natural extension of the generative exponent from Damian et al. [2024] to the multi-index setting. We show that a sample complexity of $n = \Theta(d^{1 \vee k^\star/2})$ is necessary in the class of algorithms captured by the Low-Degree-Polynomial framework; and also sufficient, by giving a sequential estimation procedure based on a spectral U-statistic over appropriate Hermite tensors.

## 1 Introduction

We consider learning Gaussian multi-index models:

**Definition 1.** *We say that $(X, Y)$ follows a Gaussian multi-index model with index $r$ if $X \sim N(0, I_d) := \gamma_d$ and there exists a subspace $U^\star \in \mathcal{G}(r, d)$ such that the conditional law $\mathbb{P}[Y|X]$ only depends on the orthogonal projection $P_{U^\star} X$.*

A Gaussian multi-index model can be thus specified by choosing a basis $W^\star$ of $U^\star$ (ie, an element of the Stiefel manifold $\mathcal{S}(r, d)$), and the law $\mathsf{P}$ of $(Z, Y) \in \mathcal{P}(\mathbb{R}^r \times \mathbb{R})$ [1], where $Z = (W^\star)^\top X$. The subspace $U^\star$ is referred as the *index space*.

Given a joint distribution $\mathsf{P}$ of $(Z, Y)$, a natural statistical task associated with such a model is to *plant* a subspace $W^\star$, uniformly drawn from the Haar measure of $\mathcal{S}(r, d)$, and draw $n$ iid samples from the multi-index distribution $\mathbb{P}_{W^\star, \mathsf{P}}$ parametrized by $W^\star$ and $\mathsf{P}$. Our task will be then to recover $U^\star = \text{span}[W^\star]$ given these samples. We note that this task is only well-posed when the 'intrinsic' dimension of the model is $r$, namely that $\mathsf{P}$ does *not* admit a factorization $\mathsf{P} = \gamma_{r'} \otimes \mathsf{P}_S$, where $\mathsf{P}_S(z_S, y)$ is the marginal of $\mathsf{P}$ over a subspace of $\mathbb{R}^r \times \mathbb{R}$ of dimension $< r + 1$ that includes the last coordinate. We will assume this property from now on.

We place ourselves in the setting where $r = O_d(1)$, and consider the high-dimensional regime. Since the dimensionality of $\mathcal{S}(r, d)$ is of order $rd$, one expects that a brute-force estimation procedure that fits $\mathbb{P}_{\mathsf{P}, W_j}$ over a suitable $\epsilon$-net of $\{W_j\}_j \subset \mathcal{S}(r, d)$ requires $O(d\epsilon^{-2})$ samples to estimate the index space up to accuracy $\epsilon$. Our main motivation is to understand this question from the lens of computational-statistical gaps: *how many samples are needed, as a function of $d, \mathsf{P}$, to produce an*

---

[1]or, more precisely, the conditional law $\mathsf{P}^Z$ of $Y|Z$, since the marginal of $Z$ is $\gamma_r$.

39th Conference on Neural Information Processing Systems (NeurIPS 2025).

*estimate of the planted subspace using polynomial-time algorithms, as opposed to using brute-force?*
This question enjoys a large literature, spanning high-dimensional statistics and learning theory, starting from the inverse regression methods from Li [1991] and beyond Xia [2008], Xia et al. [2002], Hristache et al. [2001], Cook and Li [2002], Cook [2000], Vempala [2010], Klivans et al. [2008], Mossel et al. [2003], Daniely et al. [2025] (see also Bruna and Hsu [2025] for a recent survey), where efficient algorithms have been developed for specific instances. Multi-index models are an appealing semiparametric model, and provide arguably the simplest instance of linear *feature learning*, in the sense that the index space provides an adapted low-dimensional representation to perform high-dimensional learning. Some notorious examples include

- *(noisy) Gaussian parity:* $Y|Z \overset{d}{=} \xi \operatorname{sign}[Z_1 \cdot Z_2 \ldots Z_r]$, with $\mathbb{P}[\xi = -1] = \eta$, $\mathbb{P}[\xi = 1] = 1 - \eta$ independent of $Z$ and $\eta < 1/2$.

- *Gaussian staircase functions:* $Y|Z \overset{d}{=} \phi_1(Z_1) + \phi_2(Z_1, Z_2) + \ldots + \phi_r(Z_1, \ldots, Z_r)$.

- *Intersection of $r$ half-spaces:* $Y|Z \overset{d}{=} 2 \prod_{j=1}^{r} \mathbf{1}(v_j^\top Z > \alpha_j) - 1$.

- *Low-rank shallow neural network:* $Y|Z = a^\top \rho(V^\top Z) + \xi$ for some $a \in \mathbb{R}^M, V \in \mathbb{R}^{r \times M}, \rho : \mathbb{R} \to \mathbb{R}$, additive noise $\xi$ independent of $Z$.

- *Polynomials:* $Y|Z = q(Z)$ where $q$ is a polynomial.

Focusing on the Gaussian setting, several works, starting from Dudeja and Hsu [2018], Ben Arous et al. [2021] and followed by Abbe et al. [2021, 2023], Bietti et al. [2022], Damian et al. [2022], Ba et al. [2022], Dandi et al. [2024a] have built a harmonic analysis framework to analyze a large class of algorithms, including stochastic gradient descent over NN architectures, leading to sample complexities of the form $n = \Theta(d^k)$, where $k$ is an explicit *exponent* associated with a certain harmonic expansion of P. In particular, Damian et al. [2024], focusing on Single-Index models (where $r = 1$), identified the *generative exponent* $k^\star = k^\star(\mathsf{P})$ (see Section 2) as the fundamental quantity driving the sample complexity, in the sense that $n = \Theta(d^{1 \vee k^\star/2})$ is both necessary and sufficient in the class of algorithms implemented by SQ (Statistical Queries) and Low-Degree Polynomials. In essence, the generative exponent arises from an expansion of the *inverse regression* of $Z$ given $Y$, as put forward in the original Li [1991]. Lee et al. [2024], Arnaboldi et al. [2024], Dandi et al. [2024b] showed that SGD with reused samples can learn single index models dependent on the generative exponent, instead of the information exponent.

In this work, we extend this notion of generative exponent to the general multi-index setting. As already pointed out in the literature Abbe et al. [2023], Bietti et al. [2025], Troiani et al. [2024], Diakonikolas et al. [2025b], the general $r > 1$ setting gives rise to important new phenomena not present in the single-index case. In particular, gradient-based learning exhibits a sequential behavior in the form of *saddle-to-saddle* dynamics, where the index space is revealed incrementally along specific subspaces, with different timescales associated with each step. Moreover, such incremental alignment requires solving a semi-parametric problem, where both the subspace and the link function need to be estimated jointly. We overcome these additional challenges by identifying a suitable generalization of the generative exponent, the *leap generative exponent* $k^\star$ (see Definition 3), arising from a 'canonical' orthogonal decomposition of the index space, the leap decomposition (see Section 2).

We first show that this exponent provides a computational lower bound of $n = O(d^{k^\star/2})$ under the Low-degree polynomial (LDP) framework, by extending the previously established lower bound in the single-index setting Damian et al. [2024] to an appropriate detection task that is dominated by the index estimation task (Theorem 1). Next, and more importantly, we provide an algorithm that sequentially estimates the index space along the leap decomposition from the spectrum of a novel kernel $U$-statistic (see Eq (5)). This algorithm recovers the index space as soon as $n \gtrsim d^{k^\star/2}$, thus matching the LDP lower bound, and, crucially, it does not require prior knowledge of the multi-index model P (Theorem 3). We complement these general results by several case studies that give novel guarantees on specific multi-index models, such as general ReLU networks or Gaussian Parities; see Section 5. Taken together, our results therefore provide the correct, sharp dimension dependence for *any (Gaussian) multi-index model*. In particular, as soon as $k^\star > 2$, they provide evidence of a computational-to-statistical gap at the polynomial scale.

**Related Works** Chen and Meka [2020] show that any polynomial multi-index model can be learned with $n \gtrsim d$ samples via an iterated filtered PCA algorithm. Chen et al. [2022] extended this to the case of multi-index models with ReLU activation with a similar algorithm. As we will show in Section 5, the generative exponent satisfies $k^\star \leq 2$, so our algorithm also requires only $n \gtrsim d$.

Gradient descent on two-layer networks has been extensively studied [Bietti et al., 2025, Ren and Lee, 2024, Ren et al., 2025, Damian et al., 2022, Abbe et al., 2023], these papers typically require at least $n = \Theta(d^{1 \vee l^\star - 1})$, where $l^\star$ is the information exponent [Arous et al., 2021], an upper bound of the generative exponent. Abbe et al. [2023] provide a similar definition of leap exponent but tailored to the information exponent, and thus larger than the generative leap exponent. In the setting of sparse juntas, Joshi et al. [2024] showed that by changing the loss function from square loss to another loss, gradient queries learn with complexity governed by the SQ-exponent, which is analogous to the generative exponent but restricted to juntas. Troiani et al. [2024] characterize the generative leap exponent for leaps $\leq 2$. Defilippis et al. [2025], Kovačević et al. [2025] give spectral estimators for the special case when the subspace is fully identified in the first leap of generative exponent $\leq 2$. See Section 2 for further discussion.

**Tensor PCA.** In the context of Tensor PCA, Montanari and Richard [2014] proposed the Tensor PCA model and presented several algorithms including tensor unfolding. Zheng and Tomioka [2015] proposed a rectangular unfolding algorithm closely related to a single step of our algorithm, and showed it attain the conjectured optimal sample complexity of $n = \Theta(d^{1 \vee k^\star/2})$. Dudeja and Hsu [2021] provided statistical query lower bounds for the symmetric and asymmetric Tensor PCA model, and Hopkins et al. [2015, 2017] gave the corresponding lower bound in the low-degree / SOS models. Dudeja and Hsu [2024] provided a comprehensive study of communication lower bounds and efficient algorithms for Tensor PCA and the related problem of Non-Gaussian Component Analysis. Arous et al. [2024] initiated the study of stochastic gradient descent over the Stiefel manifold for the multi-spike Tensor PCA model, showing a time complexity of $d^{1 \vee k^\star - 1}$ where $k^\star$ is the order of the tensor (analogous to the generative exponent).

Chen et al. [2020] show that deep neural networks can simulate unfolding-like algorithms and learn multi-index functions with $n = \Theta(d^{\lceil k/2 \rceil})$ where $k$ is the degree of the polynomial approximation to the groundtruth function. This method requires that the groundtruth is close to a polynomial.

Vempala [2010], Klivans et al. [2024] provides the current best known result for learning intersection of $k$-halfspaces in $d$-dimensions with $n = \Theta(d)$. Vempala and Xiao [2012] provide a moment-based algorithm for learning multi-index models when the first leap learns all relevant variables.

While this work was being finalized, we became aware of Diakonikolas et al. [2025b,a], which introduces a similar estimation procedure based on subspace conditioning. They define the class of $m$-well-behaved multi-index models. For the special case of single index models with generative exponent $k^\star$, Diakonikolas et al. [2025b][Appendix D.2] and Diakonikolas et al. [2025a][Appendix C.3.2] show $m = k^\star$; we believe a similar equivalence holds also for multi-index models. However the proposed algorithm requires sample complexity $n = d^{O(k^\star)}$ even in the realizable setting, whereas the algorithm of Damian et al. [2024] and this work, require only $n = \Theta(d^{k^\star/2})$ and apply when $y$ is either continuous or discrete. On the other hand, Diakonikolas et al. [2025b,a] algorithms aim for agnostic PAC learning, not just recovery of the subspace, and thus are able to explicitly characterize the dependence in the hidden constant $C(\mathsf{P})$ in $n \geq C(\mathsf{P})d^{1 \vee k^\star/2}$. By building an explicit piecewise constant discretization in the subspace, they explicitly characterize the dependence on $r$, $\epsilon$, and Lipschitz parameters. We expect that our subspace recovery algorithm can be combined with a discretization algorithm to attain similar guarantees, but with improved dependence on $d$. We also study several examples of the leap generative exponents in Section 5 including piecewise linear functions (deep ReLU Networks with bias) and general deep neural networks with $r$-dimensional first hidden layer, improving upon previous results specific to multi-index polynomials and homogeneous piecewise linear functions Chen and Meka [2020], Chen et al. [2022].

**Notation** $\mathbf{h}_k$ denotes the normalized $k$-th Hermite tensor, defined as $\mathbf{h}_k(u) := \frac{(-1)^k}{\sqrt{k!}} \frac{\nabla^{\otimes k} \gamma_d(u)}{\gamma_d(u)}$ for $u \in \mathbb{R}^d$. $\mathcal{S}(r, d)$ is the Stiefel manifold of $r \times d$ orthogonal matrices, and $\mathcal{G}(r, d)$ is the Grassman manifold, obtained by quotienting $\mathcal{S}(r, d)$ by $r$-dimensional basis transformations. For two subspaces

$T \subseteq T'$, we write $T' \setminus T$ as the orthogonal complement of $T$ in $T'$. For two subspaces $T, T'$ we define their distance $\mathsf{d}(T, T')$ to be $\|\Pi_T - \Pi_{T'}\|_{\mathrm{op}}$ where $\Pi_T$ is the orthogonal projection onto $T$.

**Paper outline.** Section 2 recalls the generative exponent for single-index models and shows how to generalize the definition to multi-index models via the leap generative exponent, and discuss the relation to the leap information exponent. Section 3 gives the main computational lower bound result which shows that in the low-degree polynomial framework the multi-index model with leap generative exponent $k^\star$ requires $n \geq \tilde{\Omega}(d^{k^\star/2})$. Section 4 gives our main algorithm, an iterative spectral method based on a Hermite Kernel U-statistic, that recovers the subspace with the optimal sample complexity. Finally in Section 5, we study the leap generative exponents of several function classes including piecewise linear functions and general neural networks with $r$-dimensional first hidden layer.

## 2 The Leap Decomposition

**Preliminaries: The Generative Exponent for Single-Index Models** We start by recalling the generative exponent for single-index models Damian et al. [2024]. Given $(Z, Y)$ drawn from a joint distribution $\mathsf{P} \in \mathcal{P}(\mathbb{R} \times \mathbb{R})$ with first marginal equal to a Gaussian, and such that $\mathsf{P} \neq \gamma_1 \otimes \mathsf{P}_y$, we define for each integer $k \geq 1$, $\zeta_k := \mathbb{E}[h_k(Z)|Y] \in L^2(\mathsf{P}_y)$, and $k^\star = \inf\{k; \|\zeta_k\|_{L^2(\mathsf{P}_y)} > 0\}$. Equivalently, $k^\star$ is the smallest integer $k$ such that there exists a measurable function $\mathcal{T} : \mathbb{R} \to \mathbb{R}$ and a mean-zero $k$-th degree polynomial $q$ such that $\mathbb{E}[\mathcal{T}(Y)q(Z)] \neq 0$. The main takeaway from Damian et al. [2024] is that $n = \Theta(d^{k^\star/2 \vee 1})$ is both necessary (under the SQ and the LDP frameworks) and sufficient for recovery of the planted direction [2].

**Subspace Filtration and Leap exponents:** We begin by generalizing the coefficients $\{\zeta_k\}$ from [Damian et al., 2024]. The key novel ingredient is the notion of *subspace filtration*, capturing the sequential nature of the multi-index estimation, and which appears in several existing multi-index estimation procedures Abbe et al. [2022, 2023], Bietti et al. [2025], Diakonikolas et al. [2025b]. In essence, we now need to extend the expectations $\zeta_k$, which were conditional on the label $y$, to conditional expectations on an 'augmented label' that includes all the previously estimated directions of the index space. More formally, let $S \in \mathcal{G}(r', r)$ be a subspace, and for $z \in \mathbb{R}^r$ let $z_S \in S$ denote the orthogonal projection of $z$ onto $S$. We write $\bar{y}_S := (z_S, y) \in \mathbb{R}^{1+r'}$ and $\bar{z}_S := z_{S^\perp} \in \mathbb{R}^{r-r'}$.

For any $S \in \mathcal{G}(r', r)$, we then define:

$$\zeta_{k,S} := \mathbb{E}[h_k(\bar{Z}_S)|\bar{Y}_S] \in L^2(\mathbb{R}^{1+r'}, (S^\perp)^{\otimes k}, \mathsf{P}_{\bar{Y}_S}),$$

$$\Lambda_k(S) := \mathbb{E}_{\bar{Y}_S}[\zeta_{k,S}(\bar{Y}_S) \otimes \zeta_{k,S}(\bar{Y}_S)] \in (S^\perp)^{\otimes 2k}, \quad \lambda_k^2(S) := \mathbb{E}_{\bar{Y}_S}[\|\zeta_{k,S}(\bar{Y}_S)\|_F^2].$$

Intuitively, these tensors capture whether there is any information "of order $k$" that can be captured, given knowledge of the subspace $S$. When $S = \emptyset$ and $r = 1$, these definitions reduce to those in Damian et al. [2024]. Finally, we note that these definitions only depend on the joint distribution $\mathsf{P}$ of $(Z, Y)$ and are independent of the choice of $W^\star$.

Given a subspace $S$, we define the associated null distribution $\mathsf{P}_S$ by:

$$d\mathsf{P}_S[Z, Y] = d\mathsf{P}[\bar{Z}_S]d\mathsf{P}[Z_S, Y].$$

Under $\mathsf{P}_S$, $(Y, Z_S)$ and $\bar{Z}_S$ have the same marginals as under $\mathsf{P}$, but are independent. The label transformations $\zeta_k$ appear as the Hermite coefficients of the density ratio $\frac{d\mathsf{P}}{d\mathsf{P}_S}$:

**Lemma 1** (Density Ratio expansion). *We have the following formal expansion in $L^2(\mathsf{P}_S)$:*

$$\frac{d\mathsf{P}}{d\mathsf{P}_S}[Z, Y] = \sum_{k \geq 0} \langle h_k(\bar{Z}_S), \zeta_k(Y; Z_S) \rangle.$$

This implies the following decomposition of $\chi^2(\mathsf{P}\|\mathsf{P}_S)$, whenever this divergence exists:

**Lemma 2** (Mutual Information Expansion). *If $\chi^2(\mathsf{P}\|\mathsf{P}_S) < \infty$, $\chi^2(\mathsf{P}\|\mathsf{P}_S) = \sum_{k \geq 1} \lambda_k^2(S)$.*

---

[2] and also to learn the target, by performing a subsequent dimension-free non-parametric regression.

Notice that while $\chi^2(\mathsf{P}\|\mathsf{P}_S)$ may be infinite in some cases, e.g. in deterministic models where $Y = \sigma(Z)$, the quantities $\lambda_k(S)$ are well-defined for all $k$, since $\zeta_{k,S} \in L^2(\mathsf{P}_{\bar{Y}_S})$ [3]. Given this expansion, we can immediately define the *leap* $k(S)$ of a subset $S$:

**Definition 2** (Generative Leap relative to $S$). *$k(S)$ is the smallest $k \geq 1$ such that $\lambda_k^2(S) > 0$.*

Note that $k(S) < \infty$ so long as $\mathsf{P} \neq \mathsf{P}_S$.

**The Leap Decomposition:** We will define the flag $\mathcal{F} = \{\emptyset = S_0 \subsetneq S_1 \subsetneq \cdots \subsetneq S_L = \mathbb{R}^r\}$ inductively as follows. Given a subspace $S_i$, $i \geq 0$, we define $k_{i+1} := k(S_i)$ and $S_{i+1}$ by:

$$S_{i+1} := S_i \oplus \mathrm{span}[\Lambda_{k_{i+1}}(S_i)]. \tag{1}$$

Here, we have defined the span of a symmetric tensor $T \in (\mathbb{R}^r)^{\otimes k}$ as $\mathrm{span}(T) = \mathrm{span}[\mathrm{Mat}_{r,r^{k-1}}[T]]$ where $\mathrm{Mat}_{r,r^{k-1}}[T]$ denotes $T$ reshaped as an $r \times r^{k-1}$ matrix.

**Definition 3** (Generative Leap Exponent). *Let $k_i$, $i = 1, \ldots, L$ be defined as above. The* generative leap exponent *is defined as $k^\star := \max_i k_i$.*

We now verify that the Leap decomposition is well-defined, and give a variational representation.

**Definition 4.** *Given two subspaces $S \subsetneq T$, we define the* relative leap $k(S, T)$ *of a subspace $S$ towards $T$ as*

$$k(S, T) := \inf\{k; \, T \setminus S \subseteq \oplus_{k' \leq k}\mathrm{span}(\Lambda_{k'}(S))\} . \tag{2}$$

In words, the relative leap measures the order of the Hermite tensor needed to 'reach' the subspace $T$ from conditional expectations over $y$ and $z_S$. Observe that we can relate the leaps $k(S)$ and $k(S, T)$ as $k(S) = \inf_{T; S \subsetneq T} k(S, T)$.

**Proposition 1** (Variational Characterization of Leap Generative Exponent). *The leap decomposition terminates in a finite number of steps $L \leq r$. Moreover, we have*

$$k^\star = \inf_{\mathcal{F} = \{\emptyset = R_0 \subset \cdots \subset \mathbb{R}^r\}} \max_j k(R_j, R_{j+1}) . \tag{3}$$

*Finally, $k^\star$ is invariant to rotation: if $\tilde{\mathsf{P}} = (U \otimes \mathrm{Id})_{\#}\mathsf{P}$ where $U \in \mathcal{O}_r$ is any rotation of the model, we have $k^\star(\tilde{\mathsf{P}}) = k^\star(\mathsf{P})$.*

**Relationship with Information Leap Exponent** Finally, we relate the generative leap exponent to the information leap exponent, first introduced in Abbe et al. [2023] (referred to as IsoLeap in the setting of Gaussian input data); see also Bietti et al. [2025] and Dandi et al. [2024a]. Let us first recall its definition in our context. For any $S \in \mathcal{G}(r', r)$, we define:

$$\tilde{\zeta}_{k,S} := \mathbb{E}[Y h_k(\bar{Z}_S)|Z_S] \in L^2(\mathbb{R}^{1+r'}, (S^\perp)^{\otimes k}, \mathsf{P}_{Z_S}) ,$$

$$\tilde{\Lambda}_k(S) := \mathbb{E}_{Z_S}[\tilde{\zeta}_{k,S}(Z_S) \otimes \tilde{\zeta}_{k,S}(Z_S)] \in (S^\perp)^{\otimes 2k} , \quad \tilde{\lambda}_k^2(S) := \mathbb{E}_{Z_S}\left[\left\|\tilde{\zeta}_{k,S}(Z_S)\right\|_F^2\right] .$$

By analogy with Definition 2, we define $l(S)$ to be the smallest $k$ such that $\tilde{\lambda}_k^2(S) > 0$. Equipped with this object, the information leap exponent is recovered as follows.

**Definition 5** (Information Leap Exponent, Abbe et al. [2023], Bietti et al. [2025]). *The* information leap exponent *of the multi-index model $\mathsf{P}$ is given by $l^\star := \max_i l_i$, where $l_{i+1} = l(\tilde{S}_i)$ and $(\tilde{S}_i)_i$ is defined recursively by $\tilde{S}_0 = \emptyset$ and $\tilde{S}_{i+1} = \tilde{S}_i \oplus \mathrm{span}[\tilde{\Lambda}_{l_{i+1}}(\tilde{S}_i)]$.*

Let us now relate the Information Leap exponent to the generative leap. We start with a direct generalization of [Damian et al., 2024, Prop 2.6]:

**Proposition 2** (Generative and Information Exponents relative to subspaces). *For any subspace $S$,*

$$k(S)[\mathsf{P}] = \inf_{\mathcal{T} \in L^2(\mathsf{P}_{\bar{y}_S})} l(S)[(\mathrm{Id}_z \otimes \mathcal{T}_y)_{\#}\mathsf{P}] . \tag{4}$$

*In particular we have $k(S) \leq l(S)$ for any subspace $S$.*

---

[3] one can explicitly control $\|\zeta\|$ ; see Lemma 14.

In words, the generative exponent relative to a subspace $S$ is the largest $k$ such that $\mathbb{E}_{\mathsf{P}}[\mathcal{T}(y, z_S)q(z_{S\perp})] = 0$ for any measurable function $\mathcal{T}$ and any polynomial $q$ of degree $< k$. This provides a useful characterization, as illustrated in the examples of Section 5.

As expected, the generative leap is upper bounded by the information leap:

**Proposition 3** (Relationship with Leap Information Exponent). *We have $k^\star \leq l^\star$.*

Proofs of these results are deferred to Appendix B.

## 3 Computational Lower Bounds in the Low-Degree Polynomial Class

Let us first establish a computational lower bound for the estimation of a multi-index model. Following Damian et al. [2024], and relying on the fact that detecting planted structure is a necessary byproduct of estimating the index space, we instantiate a hypothesis testing adapted to the leap decomposition.

Given $\mathsf{P}$ and its associated leap decomposition (1), we consider $\bar{S}$ the subspace of dimension $r_0$ associated with the generative leap, ie $k^\star = k(\bar{S})$. Let $\bar{y} = (\bar{S}z, y)$ be the effective label, with $\bar{y} \in \mathbb{R}^{r_0+1}$, $\bar{W} = \bar{S}^\top W^\star$ the planted subspace associated with $\bar{S}$, and $\bar{x} = \bar{W}^\perp x \in \mathbb{R}^{d-r_0}$ the effective input. Viewing $\mathsf{P}$ as the joint distribution of $(\bar{S}^\perp z, \bar{y})$, we define $\mathsf{P}_{\bar{y}}$ as the marginal over $\bar{y}$. Note that $r_0 < r$ by definition. We consider the following detection problem, *conditional* on $\bar{W}$:

- $\mathbb{H}_1$ : there is a planted model of dimension $\tilde{r} > r_0$ using $\mathsf{P}$ as link function and $\bar{y}$ as label. Specifically, $(\bar{x}, \bar{y}) \sim \mathbb{E}_{\tilde{W}} \mathbb{P}_{\tilde{W}}$, where $\mathbb{P}_{\tilde{W}}(\bar{y}|\bar{x}) = \mathsf{P}(\bar{y}|\tilde{W}^\top \bar{x})$.
- $\mathbb{H}_0$ : there is only planted structure up to dimension $r_0$; i.e., $(\bar{x}, \bar{y}) \sim \mathbb{P}_0 := \gamma_{d-r_0} \otimes \mathsf{P}_{\bar{y}}$.

By considering the likelihood ratio $\mathcal{R} = \frac{d\mathbb{H}_1}{d\mathbb{H}_0}$ and its orthogonal projection $\mathcal{R}_{\leq D}$ in $L^2(\mathbb{H}_0)$ onto polynomials of degree at most $D$, one can assess the ability of low-degree polynomials to solve this hypothesis testing problem Bandeira et al. [2022], Hopkins [2018]. Specifically, if $\|\mathcal{R}_{\leq D}\|_{L^2(\mathbb{H}_0)} = 1 + o_d(1)$, then no degree-$D$ polynomial $f$ in the input samples can weakly separate $\mathbb{H}_0$ from $\mathbb{H}_1$, ie satisfy $\max\{\text{Var}_0[f], \text{Var}_1[f]\} = O(|\mathbb{E}_0[f] - \mathbb{E}_1[f]|^2)$ as $d \to \infty$ [Bandeira et al., 2022, Proposition 6.2].

**Theorem 1** (Weak separation lower bound). *Consider $d \gg \max(r, k^\star)$, $D = O(\log(d)^2)$, and $n = O(d^{k^\star/2-\gamma})$ for any $\gamma > 0$. Then $\|\mathcal{R}_{\leq D}\|_{L^2(\mathbb{H}_0)} = 1 + o_d(1)$.*

In other words, any degree-$D$ polynomial test needs $n \geq \tilde{\Omega}(d^{k^\star/2})$ samples to weakly detect $\mathbb{H}_1$ from $\mathbb{H}_0$. Polynomial tests of degree $\omega(\log d)$ are considered a powerful step towards ruling out all noise-tolerant polynomial-time algorithms Bandeira et al. [2022], Kunisky et al. [2019]. The proof can be found in Appendix C.

This low-degree lower bound extends the previous LDP lower-bound from the single-index setting Damian et al. [2024]. In that single-index setting, this LDP lower bound agrees with a SQ lower bound of $n = \Theta(d^{k^\star/2})$ samples. While it is possible to translate our LDP lower bounds to SQ lower bounds, eg via Brennan et al. [2021], we note that there is a fundamental distinction arising in the multi-index setting, stemming from the inherent inability to perform certain spectral tasks in SQ. Dudeja and Hsu [2021] illustrated this mismatch in the setting of Tensor PCA, where asymmetric structures (such as the ones faced by multi-index model estimation) incur in additional dimension-factors. That said, some SQ lower bounds are known for the multi-index setting. Joshi et al. [2024] establishes SQ lower bounds for the number of queries of order $\Theta(d^{k^\star})$, and Diakonikolas et al. [2025b,a] obtains sample complexity lower bounds of order $\Theta(d^{k^\star/2})$ (where the generative leap is replaced by the equivalent $m$ in their notation), thus matching our LDP lower bounds.

## 4 Upper Bound via Hermite Kernel U-Statistic

We begin by describing a spectral estimator that works for a single leap. To motivate it, recall that the spectral estimator for single index models in Damian et al. [2024] began by estimating the tensor:

$$T = \mathbb{E}_{X,Y}[\mathcal{T}(Y)\boldsymbol{h}_k(x)].$$

For a suitable label transformation $\mathcal{T}$, the true expectation is proportional to $(w^\star)^{\otimes k}$, so estimating $w^\star$ is similar to a single-spike tensor PCA problem. For this problem, the partial trace estimator is an effective way to estimate $w^\star$. This estimator consists in repeatedly contracting indices $T \leftarrow T[I]$ until you are left with a vector whose expectation is $w^\star$ or a matrix whose expectation is $w^\star(w^\star)^\top$. However, this trick does not work in the multi-index setting. For example, consider Gaussian $k$-parity: $y = \mathrm{sign}(z_1 \cdots z_k)$. For this problem, we can compute the population mean of an order $k$ estimator:[4]

$$T = \mathbb{E}[Yh_k(X)] = \left(\tfrac{2}{\pi}\right)^{k/2} \sqrt{k!} \, \mathrm{Sym}(w_1^\star \otimes \cdots \otimes w_r^\star).$$

Thus, this behaves like a symmetric multi-spike tensor PCA problem. For this problem, note that because the $\{w_i^\star\}$ are mutually orthogonal, $T[I] = 0$ so taking any partial traces of this tensor will fail to produce a consistent estimator. For standard tensor PCA, this can be solved by tensor unfolding Montanari and Richard [2014]. For example, Zheng and Tomioka [2015] showed it was sufficient to unfold $T$ into a $d \times d^{k-1}$ matrix and compute the left singular vectors. Explicitly if $A = \mathrm{Mat}_{(d,d^{k-1})}[T]$ denotes $T$ reshaped as a $d \times d^{k-1}$ matrix, then you can perform a spectral decomposition of $AA^\top \in \mathbb{R}^{d \times d}$ and the top eigenvectors will recover the hidden directions.

Returning to the multi-index setting, this would motivate the following estimator. Given $n$ samples $\{(x_i, y_i)\}_{i=1}^n$, we define the embedding $\phi$, the flattened tensor $\Phi$ and the matrix estimator $M_n$ by:

$$\phi(x) := \mathrm{Mat}_{(d,d^{k-1})}[\boldsymbol{h}_k(x)], \quad \Phi = \frac{1}{n}\sum_{i=1}^n \mathcal{T}(y_i)\phi(x_i) \in \mathbb{R}^{d \times d^{k-1}}, \quad M_n := \Phi\Phi^\top \in \mathbb{R}^{d \times d}.$$

We can then perform a spectral decomposition of $M_n$. Note that this is exactly equivalent to estimating the tensor $\frac{1}{n}\sum_{i=1}^n \mathcal{T}(y_i)\boldsymbol{h}_k(x_i)$, unfolding it into a $d \times d^{k-1}$ matrix, and computing its left singular vectors. However, this strategy cannot achieve the optimal threshold of $n \gtrsim d^{\frac{k}{2}}$ because the "diagonal" terms dominate the matrix and destroy the concentration. More specifically, we can expand $M_n$ as:

$$M_n = \frac{1}{n^2}\sum_{i,j} \mathcal{T}(y_i)\mathcal{T}(y_j)\phi(x_i)\phi(x_j)^\top$$

$$= \underbrace{\frac{1}{n^2}\sum_i \mathcal{T}(y_i)^2 \phi(x_i)\phi(x_i)^\top}_{(I)} + \underbrace{\frac{1}{n^2}\sum_{i \neq j} \mathcal{T}(y_i)\mathcal{T}(y_j)\phi(x_i)\phi(x_j)^\top}_{(II)}.$$

For this estimator, one can show that the spikes in $\mathbb{E}\,M_n$ get lost in the bulk of the eigenvalues corresponding to $(I)$ unless $n \gtrsim d^{1 \vee \frac{2k-1}{3}}$, which falls short of the optimal threshold $d^{1 \vee \frac{k}{2}}$. To improve this estimator, we therefore isolate the second term $(II)$:

$$U_n = \frac{1}{n(n-1)}\sum_{i \neq j} \mathcal{T}(y_i)\mathcal{T}(y_j)\phi(x_i)\phi(x_j)^\top.$$

This is an order 2 matrix $U$-statistic which only sums over the disjoint pairs $i \neq j$. As a result the expectation is preserved: $\mathbb{E}\,U_n = \mathbb{E}\,\Phi\,\mathbb{E}\,\Phi^\top$ and we prove that $U_n$ does concentrate to its expectation in operator norm with $n \gtrsim d^{k/2}$ samples (Theorem 2).

However, it is not true in general that a single label transformation $\mathcal{T}$ is enough for $\mathbb{E}\,U_n$ to span the entire space when there are multiple leaps i.e. it may be necessary to use a label transformation $\mathcal{T}_1$ to estimate the first direction $w_1^\star$ and $\mathcal{T}_2$ to estimate $w_2^\star$. Rather than computing the top eigenvector of this matrix $U$-statistic for each label transformation $\mathcal{T}_i$, we could simply add them together into $\mathcal{T}(Y) = [\mathcal{T}_1(Y), \ldots, \mathcal{T}_m(Y)] \in \mathbb{R}^m$ and form an aggregate matrix:

$$U_n = \frac{1}{n(n-1)}\sum_{i \neq j} \phi(x_i)\phi(x_j)^\top \langle \mathcal{T}(y_i), \mathcal{T}(y_j) \rangle.$$

Because $\mathcal{T}$ only enters the $U$-statistic through inner products, we can use the kernel trick and replace it with a general PSD kernel $K$:

$$U_n = \frac{1}{n(n-1)}\sum_{i \neq j} \phi(x_i)\phi(x_j)^\top K(y_i, y_j), \tag{5}$$

---

[4]We note that because the labels lie in $\{0, 1\}$ for Gaussian parity, applying a label transformation is equivalent to an affine transformation of $T$ and therefore cannot help estimate the hidden directions.

**Algorithm 1:** A Single Leap

---

**Input:** dataset $\mathcal{D} = \{(x_i, y_i)\}_{i=1}^n$, moment $k$, recovery dimension $s$, PSD Kernel $K$
$\phi_i \leftarrow \mathrm{Mat}_{d \times d^{k-1}}[\boldsymbol{h}_k(x_i)]$ for $i = 1, \ldots, n$
$U_n \leftarrow \frac{1}{n(n-1)} \sum_{i \neq j} \phi_i \phi_j^\top K(y_i, y_j)$
$[S, V] \leftarrow \mathtt{eig}(U_n)$
**Output:** $\mathrm{span}[v_1, \ldots, v_s]$

---

which reduces to the above setting by taking $K(y_i, y_j) = \langle \mathcal{T}(y_i), \mathcal{T}(y_j) \rangle$. However, by allowing more general kernels $K$ which correspond to "infinite" embedding vectors $\mathcal{T}$, this allows to automatically average over an "infinite number" of label transformations. We will show that this allows us to learn the subspace corresponding to the next leap with the optimal sample complexity of $n \gtrsim d^{\frac{k}{2}}$ *without any knowledge of the multi-index model* $\mathsf{P}$. To begin, we prove the following lemma which controls the expectation of this matrix $U$-statistic:

**Lemma 3.** *If $K$ is integrally strictly positive definite,[5] there exist $c(\mathsf{P}, K), C(\mathsf{P}, K) > 0$ independent of $d$ such that if $S := (U^\star)^\top \mathrm{span}[\Lambda_k]$ denotes the subspace corresponding to the next leap then*

$$c(\mathsf{P}, K)\Pi_S \preceq \mathbb{E}\, U_n \preceq C(\mathsf{P}, K)\Pi_S.$$

We note that commonly used kernels like the RBF kernel automatically satisfy the assumption in Lemma 3. This implies that if we could estimate the span of $\mathbb{E}\, U_n$, we could recover the next leap. To estimate the span, we use the following theorem which bounds $U_n - \mathbb{E}\, U_n$ in operator norm:

**Theorem 2** (Concentration of U-Statistic)**.** *Let $K$ be a PSD kernel with $K(y, y) \leq 1$ for all $y$. Then if $n \gtrsim_k d^{k/2}/\epsilon + dr^k/\epsilon^2$, we have that $\|U_n - \mathbb{E}\, U_n\|_{op} \leq \epsilon$ with probability at least $1 - \exp(-d^c)$ for an absolute constant $c > 0$.*

As a corollary, by Davis-Kahan we can recover the subspace up to error $\epsilon$ with $n \gtrsim d^{k/2}/\epsilon + d/\epsilon^2$ samples where the hidden constant is independent of $d$ and depends only on the multi-index model $\mathsf{P}$:

**Corollary 1** (Subspace Recovery)**.** *For any multi-index model $\mathsf{P}$, there exists a constant $C(\mathsf{P}, K)$ independent of $d$ such that if $n \geq C(\mathsf{P}, K)\left[\frac{d^{k/2}}{\epsilon} + \frac{d}{\epsilon^2}\right]$ then the output $S \subset \mathbb{R}^d$ of Algorithm 1 satisfies $\mathsf{d}(S, (U^\star)^T \mathrm{span}[\Lambda_k]) \leq \epsilon$ with probability at least $1 - \exp(-d^c)$ for an absolute constant $c > 0$.*

### 4.1 Iterating over Leaps

Once we have recovered an partial subspace $S$, which we hope is approximately contained in $\mathrm{span}[(U^\star)^\top]$, we need to continue this process to take the next leap. We can consider the augmented label $\bar{Y}_S = (Y, \Pi_S x)$. Then $X, \bar{Y}$ again form a multi-index model with hidden dimension at most $r$ so we can repeat our matrix U-statistic estimator from the previous section. Note that the kernel $K$ now maps $\mathbb{R}^{|S|+1} \times \mathbb{R}^{|S|+1} \to \mathbb{R}$. We will denote the resulting kernel by $U_n^{(S)}$:

$$U_n^{(S)} := \frac{1}{n(n-1)} \sum_{i=1}^n \phi_i \phi_j^\top K([y_i, \Pi_S x_i], [y_j, \Pi_S x_j]).$$

We can directly apply Corollary 1 to show that for any subspace $S$, we can recover the span of $(U^\star)^T \Lambda_k(S)$ up to error $\epsilon$ with $n \gtrsim d^{k/2}/\epsilon + d/\epsilon^2$ samples. We will now control the accumulation of errors to show that we can recover the full multi-index model with $n \gtrsim C(\mathsf{P}, K)d^{k^\star/2}/\epsilon$ samples:

**Lemma 4.** *If the kernel $K$ is $L$-Lipschitz, then there exists a constant $C(\mathsf{P}, K)$ such that the map $S \to \mathbb{E}\, U_n^{(S)}$ is $C(\mathsf{P}, K)L$-Lipschitz in operator norm.*

A common example of a Lipschitz kernel is the RBF kernel which is $1/\sigma$-Lipschitz. Therefore if we run this estimator starting with the wrong subspace $\hat{S}$ with $d(S, \hat{S}) \leq \epsilon$, then the span of our estimator

---

[5]We say that $K$ is integrally strictly positive definite if for all finite non-zero signed Borel measures $\mu$, $\int K(x, y)d\mu(x)d\mu(y) > 0$. We remark that many commonly used kernels, including the RBF and Laplacian kernels, satisfy this assumption [Sriperumbudur et al., 2010].

---

**Algorithm 2:** Iterating over Leaps

---

**Input:** dataset $\mathcal{D} = \{(x_i, y_i)\}_{i=1}^n$, moments $\{k_i\}_{i=1}^m$, subspace dimensions $\{s_i\}_{i=1}^m$, Kernels $\{K_i\}_{i=1}^m$
$S \leftarrow \emptyset$
**for** $i = 1, ..., m$ **do**
    Draw $\lfloor n/m \rfloor$ fresh samples $\mathcal{D}_i$ from $\mathcal{D}$
    $y \leftarrow [y, \Pi_S x] \in \mathbb{R}^{|S|+1}$ for $(x, y) \in \mathcal{D}_i$
    $S \leftarrow S \oplus$ Algorithm 1$(\mathcal{D}_i, k_i, s_i, K_i)$
**end**
**Output:** $S$

---

can only change by $C(\mathsf{P}, K)L\epsilon$. By iterating this argument, Theorem 2 implies that Algorithm 2 will succeed in recovering $\mathrm{span}[U^{\star\top}]$ up to error $\epsilon$ given $n \gtrsim d^{k^\star/2}/\epsilon + d/\epsilon^2$ samples:

**Theorem 3** (Main Result). *For any multi-index model* $\mathsf{P}$*, there exists a constant* $C(\mathsf{P}, K)$ *independent of* $d$ *such that if* $n \geq C(\mathsf{P}, K)\left[\frac{d^{k^\star/2}}{\epsilon} + \frac{d}{\epsilon^2}\right]$ *then the output* $S \subset \mathbb{R}^d$ *of Algorithm 2 satisfies* $\mathsf{d}(S, \mathrm{span}[(U^\star)^\top]) \leq \epsilon$ *with probability at least* $1 - \exp(-d^c)$ *for some* $c = c(k^\star) > 0$.

**Remark 4.** *Our main upper bound, Algorithm 2, requires knowledge of the sizes of each leap* $\{k_i\}$ *and the dimension of each leap* $\{s_i\}$*. However, these restrictions can be easily lifted, in the spirit of Dudeja and Hsu [2018]. Using the guarantee in Theorem 2, we could start with* $k = 1$ *for each leap and increase* $k$ *until we detect outlier eigenvalues outside of the* $\sqrt{d^{k/2}/n}$*-bulk. However, for simplicity we have written the algorithm assuming knowledge of both* $\{k_i\}$ *and* $\{s_i\}$*.*

Our Algorithm 2 is thus a streamlined version of a subspace conditioned spectral method. While it shares similarities with recent methods in the literature Chen and Meka [2020], Chen et al. [2022], Diakonikolas et al. [2025b,a], Troiani et al. [2024], it crucially relies on a U-statistic in order to reach the optimal sample complexity of $d^{k^\star/2}$. An additional feature of our algorithm – that to our knowledge is novel in the literature – is the use of a generic kernel over the already discovered labels, which eliminates the need to perform successive non-parametric regressions during the subspace recovery. At the technical level, the concentration of the U-statistic is a priori challenging due to the heavy tails of the associated Hermite tensors; this is addressed using Gaussian universality results from Brailovskaya and van Handel Brailovskaya and van Handel [2024], with a dedicated analysis in the setting where $k^\star \leq 2$ to avoid spurious log-factors.

## 5 Case Studies

We conclude this article by computing the generative leap exponent of representative multi-index models. For some of these models our upper and lower bounds recover known results in the literature, but some are new. For simplicity, we focus here on noiseless models where $Y|Z = \sigma(Z)$ for a given link function $\sigma : \mathbb{R}^r \to \mathbb{R}$. Proofs for this section can be found in Appendix E.

### 5.1 Polynomial and Threshold Functions

We start by computing the generative leap for 'classic' multi-index classes given by parities, intersection of half-spaces and polynomials.

**Proposition 4** (Generative Leaps for representative models). *We have:*

    *(i)* $r$-*Gaussian Parity has* $k^\star = l^\star = r$.

    *(ii) Staircase Parity functions have* $k^\star \leq l^\star = 1$,

    *(iii) Intersection of halfspaces have* $k^\star \leq 2$,

    *(iv) Polynomials have* $k^\star \leq 2$.

For $r$-Gaussian parity, we thus obtain an efficient learning algorithm that requires $n = \Theta(d^{r/2})$ samples (which is optimal within the LDP class), and is to the best of our knowledge the first result[6]

---

[6]The agnostic improper learning algorithm of Chen et al. [2020] can potentially attain $n = \Theta(d^{\lceil r/2 \rceil})$ since the information exponent is $r$, this implies that there is a degree $r$ polynomial with non-trivial correlation with

that succeeds with $\Theta(d^{r/2})$ samples. The sample complexity of learning intersection of half-spaces is thus linear in dimension: this was known since Vempala [2010], Diakonikolas et al. [2017], Klivans et al. [2024], and for polynomials the same conclusion was established in Chen and Meka [2020]. We emphasize that while our results do capture the correct dependency in $d$, they are not fine-grained enough to provide the correct dependencies in $r$.

## 5.2 Piecewise Linear Functions

Piecewise linear continuous functions, in part motivated by ReLU architectures, have been extensively studied in the context of Gaussian Multi-index models Chen et al. [2022, 2023], Diakonikolas and Kane [2024]. When $\sigma$ is 1-homogeneous, as in bias-free ReLU networks, it is not hard to see that $k^\star \leq 2$, by considering diverging level sets $\{z; |\sigma(z)| \geq \lambda\}$ with $\lambda \to \infty$. Here we extend this result to the general piece-wise linear setting, including arbitrary ReLU networks with non-zero biases.

**Proposition 5** (Generative Leap for Piecewise Linear Functions). *If $\sigma$ is continuous and piece-wise linear then $k^\star(\sigma) \leq 2$.*

The proof exploits the analytic properties of Hermite functions, i.e. functions of the form $f(z) = p(z)\gamma(z)$. As an immediate corollary, our Algorithm from Section 4 learns arbitrary ReLU networks low-rank arbitrary ReLU networks in the proportional regime $n = \Theta(d)$: This improves the result of Chen et al. [2022] by allowing biases. Once the subspace is recovered, one could 'upgrade' to PAC learning the model using a standard non-parametric method, by regressing over the covariates $z = S^\top x$. This would incur in an additional sample complexity with potentially exponential dependencies in $r$ and $\frac{1}{\epsilon}$, but, importantly, independent of $d$.

## 5.3 Generative Leap under Linear Transformations

An important feature of the generative leap exponent is that the statement "$k^\star(\mathsf{P}) \leq k$" is an 'open' property, meaning that one should expect the leap exponent to be preserved or reduced by slightly perturbing the distribution $\mathsf{P}$. We formalize this intuition in the following result which shows that for almost all weight matrices, the generative leap is $\leq 2$.

**Proposition 6** (Generative Leap under linear transformations). *Let $\sigma(z) : \mathbb{R}^r \to \mathbb{R} \in L^2(\gamma_r)$, $\sigma \neq C$, and let $\mathcal{M}_r$ denote the set of $r \times r$ real matrices.*

* (i) *For $\Theta \in \mathcal{M}_r$, define $y_\Theta = \sigma(\Theta^\top z)$. Then $(z, y_\Theta) \sim \mathsf{P}_\Theta$ satisfies $k^\star(\mathsf{P}_\Theta) \leq 2$ for every $\Theta$, except possibly for a set of $r^2$-dimensional Lebesgue measure zero,*

* (ii) *Assume that $(z, \sigma(z)) \sim \mathsf{P}$ has a single leap with generative exponent $k^\star$. Let $\Gamma : D \subseteq \mathbb{R}^s \to \mathcal{M}_r$ be any analytic map such that $I_r \in \mathrm{Im}(\Gamma)$ and $\Gamma(\theta)$ is invertible for all $\theta \in D$. For $\theta \in D$, define $y_\theta = \sigma(\Gamma(\theta)^\top z)$. Then $(z, y_\theta) \sim \mathsf{P}_\theta$ satisfies $k^\star(\mathsf{P}_\theta) \leq k^\star$ for every $\theta$, except possibly for a set of $s$-dimensional Lebesgue measure zero.*

## 5.4 Shallow Neural Networks

Finally, we study two layer neural networks of the form $\sigma(z) = \sum_j \rho_j(z \cdot \theta_j)$. When the $\theta_j$ are orthogonal, estimating the index space requires estimating each neuron, and $k^\star(\sigma) = k^\star(\rho)$:

**Proposition 7** (Generative Leap for Orthogonal Weights). *Let $y = \sum_{j=1}^r a_j \rho(z_j)$. Then $k^\star \geq k^\star(\rho)$. Moreover, if all moments of $\rho(Z)$ exist, then $k^\star = k^\star(\rho)$.*

We can extend this result to almost all networks with unit norm, linearly independent columns:

**Corollary 2** (Non-orthogonal, invertible weights). *Let $y_V = \sum_{j=1}^r a_j \rho(v_j^\top z)$ with $\|v_j\| = 1$. Then $k^\star_V \leq k^\star(\rho)$ for all $V$, except possibly for a set of $r(r-1)$-dimensional measure $0$.*

An interesting question left for future work is whether this uniform control of the generative exponent by $k^\star(\rho)$ for any $V$ could be extended to general link functions; in other words whether the exclusion of these zero-measure sets is necessary in Proposition 6.

---

the $r$-Gaussian parity. Thus Chen et al. [2020] can be used to get error better than random guessing, but not vanishing error.

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

## A  Conclusions

In this work, we have extended the generative exponent $k^\star$ to the general class of Gaussian multi-index models, and established a tight sample complexity $n = \Theta(d^{k^\star/2\wedge1})$ for learning their associated index space under no prior knowledge of the link function. We provide a lower bound based on the low-degree polynomial framework, and a matching upper bound obtained with a novel spectral method that incrementally reveals directions of the index space from a kernel U-statistic. The resulting upper bound recovers and extends several dedicated estimation procedures for specific families of multi-index models, such as ReLU networks or intersection of half-spaces.

There are several avenues for future work. First, this paper focuses on the simple setting of isotropic Gaussian data. Extending both the information leap and generative leap to more complicated data distributions is left to future work. Next, we focus on deriving estimators that work with minimal information about the multi-index model P, and which succeed with the optimal sample complexity in the ambient dimension $d$. As a result, our sample complexity guarantees scale with constants $C(\mathsf{P})$ which could potentially be exponentially large in the hidden dimension $r$. Finally, we focus primarily on subspace estimation, as it is a requirement for full end-to-end learning.

## B  Proofs of Section 2

*Proof of Lemma 1.* This is a direct generalization of [Damian et al., 2024, Lemma D.1]. The $k$-th Hermite expansion of the likelihood ratio, viewed as a function of the label $\bar{Y}_S$, is directly

$$\mathbb{E}_{\mathsf{P}_S}\left[\frac{d\mathsf{P}}{d\mathsf{P}_S}(\bar{Z}_S,\bar{Y}_S)h_k(\bar{Z}_S)|\bar{Y}_S\right] = \mathbb{E}_{\mathsf{P}}[h_S(\bar{Z}_S)|\bar{Y}_S] = \zeta_{k,S}\,, \tag{6}$$

and thus, in $L^2(\mathsf{P}_S)$, we have

$$\frac{d\mathsf{P}}{d\mathsf{P}_S}(\bar{z}_S,\bar{y}_S) = \sum_k \langle\zeta_{k,S}(\bar{y}_S), h_k\bar{z}_S\rangle\,. \tag{7}$$

∎

*Proof of Lemma 2.* By orthogonality of Hermite polynomials:

$$\chi^2(\mathsf{P}||\mathsf{P}_S) = \mathbb{E}_{\mathsf{P}_S}\left[\frac{d\mathsf{P}}{d\mathsf{P}_S}[X,Y]^2\right] - 1 = \sum_{k\geq1}\mathbb{E}_{Y,Z_S}\left[\|\zeta_k(Y;Z_S)\|^2\right] = \sum_{k\geq1}\lambda_k^2(S).$$

∎

*Proof of Proposition 1.* The first statement follows immediately from the definition. To prove 3, we need to show that $k^\star \leq \max_j k(R_j, R_{j+1})$ for any flag $\mathcal{F}$. Let $\bar{S}$ the subspace associated with the generative leap $k^\star$, and let $j'$ be the largest index such that $R_{j'} \subseteq \bar{S}$.

We claim that for any $k$ and any pair of subspaces $T \subseteq T'$, we have $\text{span}(\Lambda_k(T)) \subseteq T' \oplus \text{span}(\Lambda_k(T'))$. Indeed, writing $Y' \in T'$ as $Y' = (Y, \tilde{Y})$ with $Y \in T$ and $\tilde{Y} \in T' \setminus T$, we have $\zeta_{k,T}(Y) = \mathbb{E}_{\tilde{Y}} \zeta_{k,T'}(Y, \tilde{Y})$ when restricted to $(T')^\perp$ (a subset of $T^\perp$). Now, suppose towards contradiction that $k_{j'} = k(R_{j'}, R_{j'+1}) < k^\star$. Since $R_{j'} \subseteq \bar{S}$, we have $\text{span}(\Lambda_k(R_{j'})) \subseteq \bar{S} \oplus \text{span}(\Lambda_k(\bar{S}))$ for $k \leq k_{j'}$. But from the definition of $k^\star$ we have $\text{span}(\Lambda_k(\bar{S})) = \emptyset$ for $k \leq k_{j'}$, which implies that $R_{j'+1} \subseteq \bar{S}$, which is a contradiction.

Finally, we verify that the generative leap is invariant to rotation by observing that $\Lambda_k$ are covariant to rotations, and therefore their associated spans preserve the same dimensions for all $k$.

∎

*Proof of Proposition 2.* The proof is an extension of [Damian et al., 2024, Prop 2.6].

To prove $k(S)[\mathsf{P}] \leq \inf_{\mathcal{T} \in L^2(\mathsf{P}_{\bar{y}_S})} l(S)[(\text{Id}_z \otimes \mathcal{T}_y)_\# \mathsf{P}]$, consider $k < k(S)$ and any $\mathcal{T} \in L^2(\mathsf{P}_{\bar{y}})$. We have

$$\mathbb{E}\left[\mathcal{T}(Y, Z_S)\mathbf{h}_k(\bar{Z}_S)|Z_S\right] = \mathbb{E}_Y\left(\mathbb{E}[\mathcal{T}(Y, Z_S)\mathbf{h}_k(\bar{Z}_S)|Y, Z_S]\right) = \mathbb{E}_Y\left(\mathcal{T}(\bar{Y}_S)\zeta_{k,S}(\bar{Y}_S)\right) = 0 , \quad (8)$$

since $\zeta_{k,S} = 0$. To prove $k(S)[\mathsf{P}] \geq \inf_{\mathcal{T} \in L^2(\mathsf{P}_{\bar{y}})} l(S)[(\text{Id}_{\bar{z}} \otimes \mathcal{T}_{\bar{y}})_\# \mathsf{P}]$, consider $\mathcal{T} = (\zeta_{k(S),S})_\beta$, where $\beta$ is a multiindex such that $(\zeta_{k(S),S})_\beta \neq 0$ (this $\beta$ must exist by definition of $k(S)$). We verify that

$$\mathbb{E}\left(\mathcal{T}(\bar{Y})H_\beta(\bar{Z})\right) = \mathbb{E}_{\bar{Y}}\left[\mathcal{T}(\bar{Y})(\zeta_{k(S),S})_\beta(\bar{Y})\right] \quad (9)$$

$$= \mathbb{E}_{\bar{Y}}\left[|(\zeta_{k(S),S})_\beta(\bar{Y})|^2\right] > 0 , \quad (10)$$

which shows that $\tilde{\zeta}_{k(S),S} \neq 0$ for the model with label transformation $\mathcal{T}$.

∎

*Proof of Proposition 3.* Consider the flag $\tilde{F} = \{\emptyset, \tilde{S}_1, \ldots, \tilde{S}_J = \mathbb{R}^r\}$ associated with the leap information exponent. We claim that $k(\tilde{S}_j, \tilde{S}_{j+1}) \leq l(\tilde{S}_j)$ for all $j \in [J]$, or equivalently that

$$\text{span}(\tilde{\Lambda}_k(\tilde{S}_j)) \subseteq \text{span}(\Lambda_k(\tilde{S}_j))$$

for $k \leq l(\tilde{S}_j)$. Indeed, observe that $\tilde{\zeta}_{k,S} = \mathbb{E}_Y[Y\zeta_{k,S}]$. As a consequence, from Proposition 1 we have that $k^\star \leq \max_j l(\tilde{S}_j) = l^\star$.

∎

# C   Proofs of Section 3

*Proof of Theorem 1.* Let $\mathcal{R}_{\tilde{W}} := \frac{d\mathbb{P}_{\tilde{W}}}{d\mathbb{P}_0}$ denote the likelihood ratio conditioned on $\tilde{W}$. We begin by computing the full likelihood ratio:

$$\mathcal{R}((x_1, y_1), \ldots, (x_n, y_n)) = \frac{\mathbb{E}_{\tilde{W}}\left[\prod_{i=1}^n \mathbb{P}_{\tilde{W}}[x_i, y_i]\right]}{\prod_{i=1}^n \mathbb{P}[x_i]\mathbb{P}[y_i]} = \mathbb{E}_{\tilde{W}}\left[\prod_{i=1}^n \mathcal{R}_W(x_i, y_i)\right].$$

Then by Lemma 1, we can expand this as

$$\mathcal{R} = \mathbb{E}_{\tilde{W}}\left[\prod_{i=1}^n \left(\sum_{k \geq 0} \langle \zeta_k(\bar{y}_i), \mathbf{h}_k(\tilde{W}^\top \bar{x}_i)\rangle\right)\right].$$

We will isolate the low degree part with respect to $\{\bar{x}_1, \ldots, \bar{x}_n\}$, which we denote by $\mathcal{R}_{\leq D}$. To compute this, we need to switch the product and the summation:

$$\mathcal{R} = \mathbb{E}_{\tilde{W}}\left[\sum_{p=0}^\infty \sum_{k_1+\ldots+k_n=p} \left(\prod_{i=1}^n \langle \zeta_{k_i}(\bar{y}_i), \mathbf{h}_{k_i}(\tilde{W}^\top \bar{x}_i)\rangle\right)\right].$$

We note that each term on the right hand side is a polynomial in $\bar{x}_1, \ldots, \bar{x}_n$ of degree $p$ which is orthogonal to all polynomials of degree less than $p$. Therefore $\mathcal{R}_{\leq D}$ is given by:

$$\mathcal{R}_{\leq D} = \mathbb{E}_{\tilde{W}}\left[\sum_{p=0}^{D} \sum_{k_1+\ldots+k_n=p} \left(\prod_{i=1}^{n} \langle \zeta_{k_i}(\bar{y}_i), \mathbf{h}_{k_i}(\tilde{W}^\top \bar{x}_i)\rangle\right)\right].$$

We can now use the orthogonality property of Hermite polynomials to compute the norms with respect to the null distribution $\mathbb{P}_0$. If if $\tilde{W}, \tilde{W}'$ are independent draws from the prior on $\tilde{W}$ then:

$$\|\mathcal{R}_{\leq D}\|_{L^2(\mathbb{P}_0)}^2 = \mathbb{E}_{\tilde{W}, \tilde{W}'}\left[\sum_{p=0}^{D} \sum_{k_1+\ldots+k_n=p} \left(\prod_{i=1}^{n} \mathbb{E}_{\mathbb{P}_0} \langle \zeta_{k_i}(\bar{y}_i) \otimes \zeta_{k_i}(\bar{y}_i), \mathbf{h}_{k_i}(\tilde{W}^\top \bar{x}_i) \otimes \mathbf{h}_{k_i}((\tilde{W}')^\top \bar{x}_i)\rangle\right)\right]$$

$$= \mathbb{E}_{\tilde{W}, \tilde{W}'}\left[\sum_{p=0}^{D} \sum_{k_1+\ldots+k_n=p} \left(\prod_{i=1}^{n} \langle \mathbb{E}[\zeta_{k_i} \otimes \zeta_{k_i}], \mathbb{E}[\mathbf{h}_{k_i}(\tilde{W}^\top \bar{x}_i) \otimes \mathbf{h}_{k_i}((\tilde{W}')^\top \bar{x}_i)]\rangle\right)\right].$$

For a pair $\Sigma, \tilde{\Sigma}$ of operators where $\Sigma$ is PSD, observe that

$$|\langle \Sigma, \tilde{\Sigma}\rangle| \leq \mathrm{Tr}\{\Sigma\}\|\tilde{\Sigma}\|_{\mathrm{op}},$$

thus

$$\|\mathcal{R}_{\leq D}\|_{L^2(\mathbb{P}_0)}^2 \leq \mathbb{E}_{\tilde{W}, \tilde{W}'}\left[\sum_{p=0}^{D} \sum_{k_1+\ldots+k_n=p} \left(\prod_{i=1}^{n} \lambda_k^2 \left\|\mathbb{E}[\mathbf{h}_{k_i}(\tilde{W}^\top \bar{x}_i) \otimes \mathbf{h}_{k_i}((\tilde{W}')^\top \bar{x}_i)]\right\|\right)\right].$$

Now, let $M = \tilde{W}^\top \tilde{W}' \in \mathbb{R}^{r \times r}$. We have the following control on the Hermite correlation term:

**Lemma 5.**

$$\left\|\mathbb{E}[\mathbf{h}_k(\tilde{W}^\top \bar{x}) \otimes \mathbf{h}_k((\tilde{W}')^\top \bar{x})]\right\|_{\mathrm{op}} = \|M\|_{\mathrm{op}}^k. \tag{11}$$

Let $z$ be a random variable with distribution $\|\tilde{W}^\top \tilde{W}'\|$, where $\tilde{W}, \tilde{W}'$ are drawn independently from the uniform prior on $\tilde{W}$, and let $\mathcal{P}_{\leq D}$ be the projection operator onto polynomials of degree at most $D$ in $z$. Note that $z$ is subgaussian satisfying $\mathbb{P}\left(b\sqrt{\frac{1}{d}} \leq z \leq a\sqrt{\frac{1}{d}}\right) \leq 1 - c_r b - \bar{c}_r e^{-a^2/4}$ for explicit constants $c_r, \bar{c}_r$; see eg [Bietti et al., 2025, Lemma 3.14]. Then we can upper bound the above expression as:

$$\|\mathcal{R}_{\leq D}\|_{L^2(\mathbb{P}_0)}^2 \leq \mathbb{E}_z\left[\mathcal{P}_{\leq D}\left[\left(\sum_{k\geq 0} \lambda_k^2 z^k\right)^n\right]\right].$$

By linearity of expectation and of the projection operator $\mathcal{P}_{\leq D}$, we can expand this using the binomial theorem:

$$\|\mathcal{R}_{\leq D}\|_{L^2(\mathbb{P}_0)}^2 \leq \sum_{j\geq 0} \binom{n}{j} \mathbb{E}\left[\mathcal{P}_{\leq D}\left[\left(\sum_{k\geq k^\star} \lambda_k^2 z^k\right)^j\right]\right].$$

We can further upper bound this expression by using that $\lambda_k^2 \leq \binom{r+k-1}{k}$ (Lemma 14). Plugging this in for $k \geq k^\star$ gives:

$$\|\mathcal{R}_{\leq D}\|_{L^2(\mathbb{P}_0)}^2 - 1 \lesssim \sum_{j=1}^{\lfloor D/k^\star \rfloor} \binom{n}{j} \mathbb{E}_z\left[\mathcal{P}_{\leq D}\left[\left(\sum_{k\geq k^\star} k^{r-1} z^k\right)^j\right]\right]$$

$$\lesssim \sum_{j=1}^{\lfloor D/k^\star \rfloor} \binom{n}{j} \mathbb{E}_z\left[\mathcal{P}_{\leq D}\left[(k^\star)^{j(r-1)} z^{jk^\star}(1-z)^{-jr}\right]\right]$$

$$\lesssim \sum_{j=1}^{\lfloor D/k^\star \rfloor} \binom{n}{j}\left[(k^\star)^{j(r-1)} \mathbb{E}_z[z^{jk^\star}]\right],$$

where the last line follows from Lemma 6. Finally, since $z$ is $\Theta(\sqrt{1/d})$-subgaussian, we have $\mathbb{E}[z^{jk^\star}] \lesssim (jk^\star/d)^{jk^\star/2}$.

Now, if $n = O(d^{k^\star/2-\gamma})$ with $\gamma > 0$ and $D = O((\log d)^2)$, we have

$$
\begin{aligned}
\|\mathcal{R}_{\leq D}\|_{L^2(\mathbb{P}_0)}^2 - 1 &\lesssim \sum_{j=1}^{\lfloor D/k^\star \rfloor} \binom{n}{j} \left[ (k^\star)^{j(r-1)} (jk^\star/d)^{jk^\star/2} \right] \\
&\lesssim n^{D/k^\star} k^{\star(r-1)D/k^\star} (D/d)^{D/2} \\
&= k^{\star(r-1)D/k^\star} (D)^{D/2} (n^{1/k^\star} d^{-1/2})^D \\
&= o_d(1) \, .
\end{aligned}
$$

∎

*Proof of Lemma 5.* Let $\mathbf{M} \in \mathbb{R}^{d^k \times d^k}$ be the matrix representation of $\mathbb{E}[\mathbf{h}_k(\tilde{W}^\top \bar{x}) \otimes \mathbf{h}_k((\tilde{W}')^\top \bar{x})]$. Let $\mathcal{H}_k \subset L^2(\mathbb{R}^r, \gamma)$ be the space spanned by harmonics of degree $k$. Observe that for $f, \tilde{f} \in \mathcal{H}_k$, $f = \sum_{|\beta|=k} c_\beta h_\beta$, $\tilde{f} = \sum_{|\beta|=k} \tilde{c}_\beta h_\beta$ with $c_\beta = \langle f, h_\beta \rangle, \tilde{c}_\beta = \langle \tilde{f}, h_\beta \rangle$ we have

$$
c^\top \mathbf{M} \tilde{c} = \langle P_W f, P_{W'} \tilde{f} \rangle_{\gamma_d} \, , \tag{12}
$$

where $P_W f(x) = f(W^\top x)$. We deduce that $\mathbf{M}$ is the 'averaging operator' $\mathsf{A}_M$ from [Bietti et al., 2025, Definition 1.1], restricted at harmonic $k$. From the SVD of $M = U\Lambda V^\top$, we have [Bietti et al., 2025, Corollary 2.8] that

$$
\mathbf{M} = \sum_{|\beta|=k} \lambda^\beta H_\beta(U) \otimes H_\beta(V) \, , \tag{13}
$$

with $\lambda^\beta = \prod_j \lambda_j^{\beta_j}$. We thus conclude that $\|\mathbf{M}\|_{\mathrm{op}} = \lambda_{\max}^k = \|M\|^k$. ∎

**Lemma 6.** *Let $z = \|W^\top W'\|_{\mathrm{op}}$, where $W, W'$ are drawn iid from the Haar measure of $\mathcal{S}(r,d)$. Then, for $l, \tilde{l} \leq d/4$, we have*

$$
\mathbb{E}_z \left[ z^l (1-z)^{-\tilde{l}} \right] \lesssim \mathbb{E}_z \left[ z^l \right] \, . \tag{14}
$$

*Proof.* The proof is adapted from [Damian et al., 2023, Lemma 26] to the $r > 1$ setting. From [Bietti et al., 2025, Eq (197)], the joint distribution of singular values $0 \leq \lambda_r \leq \lambda_{r-1} \cdots \leq \lambda_1$ of $M$ is given by

$$
p_{r,d}(\lambda_1, \ldots, \lambda_r) = Z_{r,d}^{-1} \prod_{i<j} (\lambda_i^2 - \lambda_j^2) \prod_{i=1}^{r} (1-\lambda_i^2)^{(d-2r-1)/2} \mathbf{1}(0 \leq \lambda_r \leq \cdots \leq \lambda_1 \leq 1) \, , \tag{15}
$$

with $Z_{r,d} = \frac{\Gamma_r^2(r/2)}{\pi^{r^2/2}} \frac{\Gamma_r((d-r)/2)}{\Gamma_r(d/2)}$. We have

$$
\mathbb{E} \left( z^l (1-z)^{-\tilde{l}} \right) = \int \lambda_1^l (1-\lambda_1)^{-\tilde{l}} p_{r,d}(\lambda_1, \ldots, \lambda_r) d\lambda_1 \ldots d\lambda_r \tag{16}
$$

From $\lambda_1 \leq 1$ we have $1 - \lambda_1^2 \leq 2(1-\lambda_1)$ so $(1-\lambda_1)^{-\tilde{l}} \leq 2^{\tilde{l}}(1-\lambda_1^2)^{-\tilde{l}}$, thus

$$
\mathbb{E} \left( z^l (1-z)^{-\tilde{l}} \right) \leq 2^{\tilde{l}} \int \lambda_1^l (1-\lambda_1^2)^{-\tilde{l}} p_{r,d}(\lambda_1, \ldots, \lambda_r) d\lambda_1 \ldots d\lambda_r \tag{17}
$$

$$
= 2^{\tilde{l}} Z_{r,d}^{-1} \int \lambda_1^l \prod_{i<j} (\lambda_i^2 - \lambda_j^2) \prod_{i=1}^{r} (1-\lambda_i^2)^{(d-2r-1-2\tilde{l})/2} \mathbf{1}(0 \leq \lambda_r \leq \cdots \leq \lambda_1 \leq 1) d\lambda_1 \ldots d\lambda_r
$$

$$
\tag{18}
$$

$$
= 2^{\tilde{l}} \frac{Z_{r,d-2\tilde{l}}}{Z_{r,d}} \mathbb{E}_{\tilde{z}}[\tilde{z}^l] \, , \tag{19}
$$

where $\tilde{z}$ is the largest singular value of $M = W^\top W'$ with $W, W' \in \mathcal{S}(r, d - 2\tilde{l})$. For $d \gg 1$, we thus conclude that $\mathbb{E} \left( z^l (1-z)^{-\tilde{l}} \right) \lesssim \mathbb{E}_z[z^l]$. ∎

# D  Proofs of Section 4

**Lemma 3.** *If $K$ is integrally strictly positive definite,[7] there exist $c(\mathsf{P}, K), C(\mathsf{P}, K) > 0$ independent of $d$ such that if $S := (U^\star)^\top \mathrm{span}[\Lambda_k]$ denotes the subspace corresponding to the next leap then*

$$c(\mathsf{P}, K)\Pi_S \preceq \mathbb{E}\, U_n \preceq C(\mathsf{P}, K)\Pi_S.$$

*Proof.* Let $\mathcal{K}$ be the kernel operator:

$$(\mathcal{K}f)(y) = \mathbb{E}_Y[K(Y, y)f(y)].$$

Using that $\langle \mathbb{E}[\boldsymbol{h}_k(X)|Y], v^{\otimes k}\rangle = \langle \zeta_k(Y), u^{\otimes k}\rangle$ where $u = U^\star v \in \mathbb{R}^r$, we have that for any $v$:

$$v^\top \mathbb{E}\, M_n v = u^\top \mathbb{E}\big[\mathrm{Mat}_{(1, k-1)}[\zeta_k(Y)]\,\mathrm{Mat}_{(1, k-1)}[\zeta_k(Y')]^\top K(Y, Y')\big] u$$
$$= \langle \zeta_k(\cdot)[u], \mathcal{K}\zeta_k(\cdot)[u]\rangle.$$

First, because $K(y, y) \leq 1$ we have that $\|\mathcal{K}\|_{op} \leq 1$ so this is upper bounded by

$$\mathbb{E}_Y \|\zeta_k(Y)[u]\|^2 \leq \lambda_k^2 \|u\|^2.$$

Therefore $\mathbb{E}\, M_n \preceq C(\mathsf{P}, K)\Pi_S$ with $C(\mathsf{P}, K) = \lambda_k^2$. Next, let $v \in S$ with $\|v\| = 1$ so that $\zeta_k(Y)[v] \neq 0 \in L^2(\mathsf{P}_y)$. Then because $K$ is injective we have that

$$c(v) := \langle \zeta_k(\cdot)[v], \mathcal{K}\zeta_k(\cdot)[v]\rangle > 0.$$

Therefore by compactness, if $C(\mathsf{P}, K)$ denotes the minimum value of $c(v)$ over the unit vectors in $S$, we have that $C(\mathsf{P}, K) > 0$. In addition we have that $\mathbb{E}\, M_n \succeq C(\mathsf{P}, K)\Pi_S$ which completes the proof. ∎

**Theorem 2** (Concentration of U-Statistic). *Let $K$ be a PSD kernel with $K(y, y) \leq 1$ for all $y$. Then if $n \gtrsim_k d^{k/2}/\epsilon + dr^k/\epsilon^2$, we have that $\|U_n - \mathbb{E}\, U_n\|_{op} \leq \epsilon$ with probability at least $1 - \exp(-d^c)$ for an absolute constant $c > 0$.*

*Proof.* The cases $k = 1, 2$ are deffered to Proposition 8 so we will assume that $k > 2$. Note that by the standard decoupling argument ([de la Peña and Giné, 1999, Theorem 3.4.1]), it suffices to control the tails of the decoupled $U$-statistic:

$$M_n := \frac{1}{n(n-1)} \sum_{i \neq j} \phi(x_i)\phi(x_j')^T K(y_i, y_j')$$

where $\{(x_i', y_i')\}_{i=1}^n$ are an i.i.d. copy of $\{(x_i, y_i)\}_{i=1}^n$. We will begin by applying Corollary 3 with respect to the randomness in $\{(x_i, y_i)\}_{i=1}^n$, treating the replicas $\{(x_i', y_i')\}_{i=1}^n$ as fixed. Define

$$V_i'(Y) := \frac{1}{n-1} \sum_{j \neq i} \phi(x_j')K(Y, y_j')$$

so that

$$M_n = \frac{1}{n} \sum_i \phi(x_i)V_i'(y_i)^T = \frac{1}{n} \sum_i Z_i$$

where $Z_i = \phi(x_i)V_i'(y_i)^T$. Then:

$$\|Z_i\|_{op}^2 \leq \|Z_i\|_F^2 = \sum_{a,b=1}^d \langle \phi(X)^T e_a, V_i'(Y)^T e_b\rangle^2.$$

---

[7]We say that $K$ is integrally strictly positive definite if for all finite non-zero signed Borel measures $\mu$, $\int K(x, y)d\mu(x)d\mu(y) > 0$. We remark that many commonly used kernels, including the RBF and Laplacian kernels, satisfy this assumption [Sriperumbudur et al., 2010].

Taking $p/2$ norms and using Lemma 9 gives for $p \geq r \log r$:

$$\left\| \|Z_i\|_{op} \right\|_p^2 = \left\| \|Z_i\|_{op}^2 \right\|_{p/2}$$

$$\leq \sum_{a,b=1}^d \left\| \langle \phi(X)^T e_a, V_i'(Y)^T e_b \rangle^2 \right\|_{p/2}$$

$$= \sum_{a,b=1}^d \left\| \langle \phi(X)^T e_a, V_i'(Y)^T e_b \rangle \right\|_p^2$$

$$\lesssim_k p^k \sum_{a,b=1}^d \left\| \|V_n'(Y)^T e_b\|_F \right\|_{2p}^2$$

$$\leq p^k d^2 \left\| \|V_n'(Y)\|_{op} \right\|_{2p}^2.$$

Next we will compute $\sigma_*(Z_i)$:

$$\sigma_*(Z_i)^2 = \sup_{\|u\|=\|v\|=1} \mathbb{E}\left[ \langle \phi^T u, V_i'(Y)^T v \rangle^2 \right] \lesssim_k \left\| \|V_i'(Y)\|_{op} \right\|_4^2$$

by the same argument as above. Therefore applying Corollary 3 gives that if $c = \frac{k-2}{2(k+4)} \geq \frac{1}{14}$ then for $p \leq d^c$,

$$\left\| \|M_n - \mathbb{E}\, M_n\|_{op} \right\|_p \lesssim \left\| \|V_i'(Y)\|_{op} \right\|_{2p} \sqrt{\frac{d}{n}}.$$

Now let $\mathbb{E}'$ denote the expectation with respect to the replicas $\{(x_i', y_i')\}_{i=1}^n$. Then by Corollary 3:

$$(\mathbb{E}'\, \mathbb{E}\, \|U - \mathbb{E}\, U\|_{op}^p)^{1/p} \lesssim \sup_i \left( \mathbb{E}' \left\| \|V_i'(Y)\|_{op} \right\|_{2p}^p \right)^{1/p} \sqrt{\frac{d}{n}} = \sup_i \left( \mathbb{E}_Y\, \mathbb{E}' \|V_i'(Y)\|_{op}^{2p} \right)^{\frac{1}{2p}} \sqrt{\frac{d}{n}}.$$

Now we decompose:

$$\|V_i'(Y)\|_{op} \leq \left\| \mathbb{E}'\, V_i'(Y) \right\| + \|V_i'(Y) - \mathbb{E}\, V_i'(Y)\|.$$

Because $|K(Y, y_j')| \leq 1$, we can use Lemma 8 and a standard symmetrization argument to show that the second term has $p$-norms bounded by $O(\sqrt{\max(d, d^{k-1})/n})$ for $p < d^c$. For the first term we have

$$\left\| \mathbb{E}'\, V_i'(Y) \right\|_{op} \leq \left\| \mathbb{E}'\, V_i'(Y) \right\|_F = \|\mathbb{E}_{Y'}\, \zeta_k(Y')K(Y, Y')\|_F \leq \sqrt{\mathbb{E}_{Y'}[\|\zeta_k(Y')\|_F^2]} \leq r^{k/2}.$$

Combining everything and applying Markov's inequality gives that with probability at least $1 -$ poly$(n)e^{-d^c}$,

$$\|U_n - \mathbb{E}\, U_n\|_{op} \lesssim \left[ r^{k/2} + \sqrt{\frac{\max(d, d^{k-1})}{n}} \right] \sqrt{\frac{d}{n}} \lesssim \frac{d^{k/2}}{n} + r^{k/2}\sqrt{\frac{d}{n}}.$$

$\blacksquare$

**Proposition 8.** *If $k \leq 2$ and $n \geq d$, we have with probability at least $1 - 2ne^{-d}$,*

$$\|U_n - \mathbb{E}\, U_n\|_{op} \leq r^{k/2}\sqrt{\frac{d}{n}}.$$

*Proof.* We can use [de la Peña and Giné, 1999, Theorem 3.4.1] to reduce the problem to concentrating:

$$M_n := \frac{1}{n(n-1)} \sum_{i \neq j} \phi_i {\phi_j'}^T K(y_i, y_j').$$

where $\mathcal{D}' = \{(x'_i, y'_i)\}_{i=1}^n$ are an i.i.d. copy of $\mathcal{D} = \{(x_i, y_i)\}_{i=1}^n$. If we define $V'_i(y) := \frac{1}{n-1} \sum_{j \neq i} \phi'_j K(y, y'_j)$, we can rewrite this as:

$$M_n = \frac{1}{n} \sum_i \phi_i V'_i(y_i)^T.$$

Now let $R$ be a truncation radius to be chosen later and let $\rho_i := \mathbf{1}_{\|V'_i(y_i)\|_{op} \leq R}$. Then define:

$$\widetilde{M}_n = \frac{1}{n} \sum_i \boldsymbol{h}_2(x_i) V'_i(y_i) \rho_i.$$

First, we have that for any unit vectors $u, v$ and any $p \geq r^2$,

$$\mathbb{E}_{\mathcal{D}} \left[ (u^T \phi_i V'_i(y_i)^T v)^p \rho_i \right]^{1/p} \leq (2p-1) \binom{r+2p}{r}^{\frac{1}{2p}} \mathbb{E}_{\mathcal{D}}[(u^T V'_i(y_i)^T v)^{2p} \rho_i]^{\frac{1}{2p}} \lesssim Rp.$$

Therefore the summands in $u^T \widetilde{M}_n v$ are subexponential so by Bernstein's inequality we have that with probability at least $1 - \delta$ over the randomness in $\mathcal{D}$,

$$\left| u^T [\widetilde{M}_n - \mathbb{E}\, \widetilde{M}_n] v \right| \lesssim R \left[ \sqrt{\frac{\log(2/\delta)}{n}} + \frac{\log(2/\delta)}{n} \right].$$

We can now union bound over a $1/4$-net of $S^{d-1}$ to get that with probability at least $1 - \delta$ over $\mathcal{D}$,

$$\left\| \widetilde{M}_n - \mathbb{E}\, \widetilde{M}_n \right\| \lesssim R \left[ \sqrt{\frac{d + \log(2/\delta)}{n}} + \frac{d + \log(2/\delta)}{n} \right].$$

Taking $\delta = e^{-cd}$ and using $n \geq d$ gives that with probability at least $1 - e^{-cd}$,

$$\left\| \widetilde{M}_n - \mathbb{E}\, \widetilde{M}_n \right\| \lesssim R \sqrt{\frac{d}{n}}.$$

Next, note that for any fixed unit vectors $u, v$ (possibly degenerate if $k = 1$), $u^T \phi'_j v K(y, y'_j)$ is a sub-exponential random variable so by Bernstein's inequality we have with probability at least $1 - \delta$ over the randomness in $\{(x'_i, y'_i)\}$,

$$\left| u^T (V'_i(y) - \mathbb{E}\, V'_i(y)) v \right| \lesssim \sqrt{\frac{\log(2/\delta)}{n}} + \frac{\log(2/\delta)}{n}.$$

Taking a union bound over a $1/4$-net of $S^{d-1}$ gives that with probability at least $1 - \delta$ over the randomness in $\mathcal{D}'$,

$$\|V'_i(y) - \mathbb{E}\, V'_i(y)\|_{op} \lesssim \sqrt{\frac{d + \log(2/\delta)}{n}} + \frac{d + \log(2/\delta)}{n}.$$

In addition, as in the proof of Theorem 2, we have that $\|\mathbb{E}\, V'_i(y)\|_{op} \lesssim r^{k/2}$. Therefore if we take $R = C r^{k/2}$ for a sufficiently large constant $C$, we have that with probability at least $1 - e^{-cd}$ that $\|V_i(Y)\|_{op} \leq R$. Therefore with probability at least $1 - ne^{-cd}$, $\widetilde{M}_n = M_n$. Furthermore, if $1 - ne^{-cd} > 0$ so that the theorem is not vacuous,

$$
\begin{aligned}
u^T \left[ \mathbb{E}\, \widetilde{M}_n - \mathbb{E}\, M_n \right] v &= \frac{1}{n} \sum_{i=1}^n \mathbb{E}\left[ u^T \phi_i V'(y_i)^T v (1 - \rho_i) \right] \\
&\leq \frac{1}{n} \sqrt{ \sum_{i=1}^n \mathbb{E}\left[ (u^T \phi_i V'(y_i)^T v)^2 \right] \sum_{i=1}^n \mathbb{P}[\|V'_i(y_i)\| \geq R] } \qquad \text{(Cauchy)} \\
&\leq r^{k/2} e^{-cd/2} \\
&\leq \frac{r^{k/2}}{\sqrt{n}}. \qquad\qquad\qquad\qquad\qquad\qquad\qquad\qquad (ne^{-cd} < 1)
\end{aligned}
$$

Putting everything together gives that with probability at least $1 - 2ne^{-cd}$,

$$\|M_n - \mathbb{E}\, M_n\|_{op} \lesssim r^{k/2}\sqrt{\frac{d}{n}}.$$

$\blacksquare$

**Lemma 4.** *If the kernel $K$ is $L$-Lipschitz, then there exists a constant $C(\mathsf{P}, K)$ such that the map $S \to \mathbb{E}\, U_n^{(S)}$ is $C(\mathsf{P}, K)L$-Lipschitz in operator norm.*

*Proof.* Let $Z(Y; X) := \mathrm{Mat}_{(1,k)}[\zeta_k(Y; X_{S \oplus S'})]$. Then,

$$\left\|\mathbb{E}\, U_n^{(S)} - \mathbb{E}\, U_n^{(S')}\right\|_F = \left\|\mathbb{E}_{X,Y}\, Z(Y; X)Z(Y'; X')^T E(Y, X)\right\|_F$$

where

$$E(Y, X) := K((Y, X_S), (Y', X'_S)) - K((Y, X_{S'}), (Y', X'_{S'})).$$

Note that

$$E(Y, X) \le L\sqrt{\|X_S - X_{S'}\|^2 + \|X'_S - X'_{S'}\|^2} \le 2d(S, S')L\max(\|X\|, \|X'\|).$$

Then by Holder's inequality,

$$\begin{aligned}
\left\|\mathbb{E}\, U_n^{(S)} - \mathbb{E}\, U_n^{(S')}\right\|_F &\le \|\|Z(Y; X)\|_F\|_4^2\, \|E(Y, X)\|_2 \\
&\le 2(3r)^k \sqrt{r} L d(S, S') \\
&\lesssim_k r^{k+1} L d(S, S').
\end{aligned}$$

$\blacksquare$

**Theorem 3** (Main Result). *For any multi-index model $\mathsf{P}$, there exists a constant $C(\mathsf{P}, K)$ independent of $d$ such that if $n \ge C(\mathsf{P}, K)\left[\frac{d^{k^\star/2}}{\epsilon} + \frac{d}{\epsilon^2}\right]$ then the output $S \subset \mathbb{R}^d$ of Algorithm 2 satisfies $\mathsf{d}(S, \mathrm{span}[(U^\star)^\top]) \le \epsilon$ with probability at least $1 - \exp(-d^c)$ for some $c = c(k^\star) > 0$.*

*Proof.* Recall the leap decomposition $\mathcal{F} = \{\emptyset = S_0^\star \subsetneq S_1^\star \subsetneq \cdots \subsetneq S_L^\star = \mathbb{R}^r\}$. We will prove by induction that for any $\epsilon > 0$, there exists a constant $C(\mathsf{P}, K)$ such that the output $S_i$ of Algorithm 2 at step $i$ satisfies $d(S_i, (U^\star)^T S_i^\star)$ with high probability whenever

$$n \ge C(\mathsf{P}, K)\left[\frac{d^{k/2}}{\epsilon} + \frac{d}{\epsilon^2}\right].$$

Note that for $i = 1$ the result is implied directly by Corollary 1. Now assume the result for $i > 1$. By Lemma 3

$$\mathbb{E}\, U_n^{(U^\star)^T S_i^\star} \succeq c(\mathsf{P}, K)\Pi_{(U^\star)^T S_{i+1}}.$$

In addition we have by Theorem 2,

$$\left\|U_n^{(S_i)} - \mathbb{E}\, U_n^{(S_i)}\right\|_{op} \lesssim_k \frac{d^{k/2}}{n} + r^{k/2}\sqrt{\frac{d}{n}}.$$

Finally by Lemma 4,

$$\left\|\mathbb{E}\, U_n^{S_i} - \mathbb{E}\, U_n^{(U^\star)^T S_i^\star}\right\|_{op} \lesssim_k r^{k+1} \times L \times d(S_i, (U^\star)^T S_i^\star).$$

Putting it all together we have that:

$$\left\|U_n^{(S_i)} - \mathbb{E}\, U_n^{(U^\star)^T S_i^\star}\right\| \lesssim_k \frac{d^{k/2}}{n} + r^{k/2}\sqrt{\frac{d}{n}} + r^{k+1} \times L \times d(S_i, (U^\star)^T S_i^\star).$$

In addition, by the induction hypothesis, $d(S_i, (U^\star)^T S_i^\star) \le C(\mathsf{P}, K)\left[\frac{d^{k/2}}{n} + r^{k/2}\sqrt{\frac{d}{n}}\right]$. Therefore,

$$\left\|U_n^{(S_i)} - \mathbb{E}\, U_n^{(U^\star)^T S_i^\star}\right\| \lesssim_k C(\mathsf{P}, K)\left[\frac{d^{k/2}}{n} + r^{k/2}\sqrt{\frac{d}{n}}\right]$$

and the result again follows from the Davis-Kahan inequality. $\blacksquare$

## D.1 Auxiliary Lemmas for Concentration

We will start with this simple inequality on the Frobenius norm of a Hermite tensor:

**Lemma 7.** $\|\boldsymbol{h}_k(X)\|_F \lesssim_k \|X\|^k + d^{k/4}.$

*Proof.* We will use the identity:

$$\boldsymbol{h}_k(X) = \frac{1}{\sqrt{k!}}\mathbb{E}_{Z\sim N(0,I_d)}[(X+iW)^{\otimes k}].$$

Therefore,

$$
\begin{aligned}
\|\boldsymbol{h}_k(X)\|_F^2 &= \frac{1}{k!}\mathbb{E}_{Z,Z'}\left[((X+iZ)\cdot(X+iZ'))^k\right]\\
&= \frac{1}{k!}\mathbb{E}_{Z,Z'}\left[(\|X\|^2 - Z\cdot Z' + iX\cdot(Z+Z'))^k\right]\\
&\leq \frac{3^{k-1}}{k!}\left[\|X\|^{2k} + \mathbb{E}_{Z,Z'}[|Z\cdot Z'|^k] + \mathbb{E}_{Z,Z'}[|X\cdot(Z+Z')|^k]\right]\\
&\lesssim_k \|X\|^{2k} + d^{k/2} + \|X\|^k\\
&\lesssim_k \|X\|^{2k} + d^{k/2}.
\end{aligned}
$$

■

We can use this to concentrate sums of $\sum_{i=1}^n c_i\phi(x_i)$ in operator norm:

**Lemma 8.** *There exists an absolute constant $C_k$ such that if $n = d^{1+\epsilon}$ with $\epsilon > 0$ and for any constants $c_i$ with $|c_i| \leq 1$ and $p = d^c$ where $c = \min(1, \epsilon/4)$,*

$$\mathbb{E}\left[\left\|\frac{1}{n}\sum_{i=1}^n c_i\phi(x_i)\right\|_{op}^p\right]^{1/p} \leq C_k\sqrt{\frac{\max(d, d^{k-1})}{n}}.$$

*Proof.* Note that for $k = 1, 2$ this follows from the standard bounds for a Gaussian covariance matrix ($k = 2$) and the norm of a Gaussian vector ($k = 1$). Therefore we will assume $k > 2$. We will begin by computing $\sigma_*(\phi)$:

$$\sigma_*(\phi) = \sup_{\|u\|=\|v\|=1}\mathbb{E}[(u^T\phi v)^2] = \mathbb{E}\langle\mathrm{vec}[\boldsymbol{h}_k(X)], \mathrm{vec}[u\otimes v]\rangle^2 \leq 1.$$

Next, $\|\phi\|_{op} \leq \|\phi\|_F \lesssim_k \|X\|^k + d^{k/4}$. Therefore the $p$-norms of $\|\phi\|_{op}$ are bounded by $d^{k/2}$ for any $p \leq d$. Plugging this into Lemma 12 gives that for $p \leq d$,

$$\left\|\left\|\frac{1}{n}\sum_{i=1}^n c_i\phi(x_i)\right\|_{op}\right\|_p \lesssim \sqrt{\frac{d^{k-1}}{n}} + \left(\frac{d^{k/2}}{n}\right)^{1/3}\left(\frac{d^{k-1}}{n}\right)^{1/3}p^{2/3} + \frac{d^{k/2}p}{n}.$$

Plugging in $p = d^c$ gives that the second and third terms are dominated by the first which completes the proof. ■

We will also use the following simple lemma:

**Lemma 9.** *Let $(X, Y)$ follow a Gaussian multi-index model with hidden dimension $r$. Then for any $k$-tensor-valued random variable $F(Y)$,*

$$\|\langle\boldsymbol{h}_k(X), F(Y)\rangle\|_p \leq (2p-1)^{\frac{k}{2}}\binom{r+kp}{r}^{\frac{1}{2p}}\|\|F(Y)\|_F\|_{2p}.$$

*Proof.* Without loss of generality we can assume $p$ is even. Now $\langle\boldsymbol{h}_k(X), f(Y)\rangle^p$ is a polynomial of degree $kp$ in $X$. Therefore by Lemma 15,

$$\mathbb{E}\langle\boldsymbol{h}_k(X), F(Y)\rangle^p \leq \sqrt{\mathbb{E}_0\langle\boldsymbol{h}_k(X), F(Y)\rangle^{2p}\binom{r+kp}{r}} \leq (2p-1)^{\frac{kp}{2}}\sqrt{\binom{r+kp}{r}\mathbb{E}_Y\|F(Y)\|^{2p}}.$$

Taking $p$th roots gives:

$$\| \langle \boldsymbol{h}_k(X), F(Y) \rangle \|_p \leq (2p-1)^{\frac{k}{2}} \binom{r+kp}{r}^{\frac{1}{2p}} \| \| F(Y) \|_F \|_{2p} .$$

$\blacksquare$

**Lemma 10** (Gaussian hypercontractivity)**.** *Let $f$ be a polynomial of degree $k$ and let $X \sim N(0, I_d)$. Then for $p \geq 2$,*

$$\mathbb{E}_X[|f(X)|^p]^{2/p} \leq (p-1)^k \, \mathbb{E}_X[f(X)^2].$$

**Lemma 11.** *Let $X, Y$ be random variables with $\|Y\|_p \leq B p^{k/2}$ for*

$$p = \min \left( 2, \frac{1}{k} \cdot \log \left( \frac{\|X\|_2}{\|X\|_1} \right) \right).$$

*Then,*

$$\mathbb{E}[XY] \leq \|X\|_1 \cdot B \cdot (ep)^{k/2}.$$

For any mean zero random matrix $Y$ we define:

$$\sigma(Y) := \max \left( \left\| \mathbb{E}[YY^\top] \right\|_2 , \left\| \mathbb{E}[Y^\top Y] \right\|_2 \right)^{1/2}$$
$$\sigma_*(Y) := \sup_{\|u\|=\|v\|=1} \mathbb{E}[(u^\top Y v)^2]$$

For non-centered matrices, we define $\sigma(Y) := \sigma(Y - \mathbb{E}\,Y)$ and $\sigma_*(Y) := \sigma_*(Y - \mathbb{E}\,Y)$.

We will rely on the following simple corollary of [Brailovskaya and van Handel, 2024, Theorem 2.6]:

**Lemma 12.** *Let $Y = \sum_{i=1}^n Z_i$ where $Z_i$ are mean zero independent random matrices. Assume that for all $i$, $\mathbb{P}[\|Z_i\| > R] \leq \delta$. Then there exists an absolute constant $C$ such that for any $t \geq 0$, with probability at least $1 - n\delta - de^{-t}$,*

$$\|Y\| \leq C \Big[ \sigma(Y) + \sigma_*(Y) t^{1/2} + R^{1/3} \sigma(Y)^{2/3} t^{2/3} + Rt \Big].$$

*Proof.* Define $\tilde{Z}_i := Z_i \mathbf{1}_{\|Z_i\|_2 \leq R}$ and let $\tilde{Y} := \sum_{i=1}^n \tilde{Z}_i$. Then,

$$\sigma(\tilde{Y}) = n^{1/2} \sigma(\tilde{Z}) \leq n^{1/2} \sigma(Z) = \sigma(Y)$$

and similarly for $\sigma_*$. In addition, by definition, $\left\| \tilde{Z}_i \right\| \leq R$. Therefore, by [Brailovskaya and van Handel, 2024, Theorem 2.6] and [Bandeira et al., 2023, Lemma 4.10], there exists a constant $C$ such that for any $t \geq 0$, with probability at least $1 - de^{-t}$,

$$\left\| \tilde{Y} - \mathbb{E}\,\tilde{Y} \right\| \leq C \Big[ \sigma(Y) + \sigma_*(Y) t^{1/2} + R^{1/3} \sigma(Y)^{2/3} t^{2/3} + Rt \Big].$$

Next, note that

$$\left\| \mathbb{E}\,Y - \mathbb{E}\,\tilde{Y} \right\|_{op} = \left\| \sum_i \mathbb{E}[Z_i \mathbf{1}_{\|Z_i\|_2 > R}] \right\| \leq \sum_i \sigma_*(Z_i) \sqrt{\delta} \leq \sigma_*(Y) \sqrt{n\delta}.$$

Now if $\delta > 1/n$, then $1 - n\delta < 0$ so the result is trivially true. Otherwise, $\| \mathbb{E}\,Y - \mathbb{E}\,\tilde{Y} \|_{op} \leq \sigma_*(Y)$. Finally, as $\tilde{Y} = Y$ on the event that $\max_i \|Z_i\| \leq R$, a union bound completes the proof. $\blacksquare$

We will use the following simple lemmas about $\sigma, \sigma_*$:

**Lemma 13.** *For any random matrix $A \in \mathbb{R}^{d \times s}$, $\sigma(A)^2 \leq \max(d, s) \sigma_*(A)^2$.*

*Proof.* Without loss of generality we can assume $\mathbb{E}[A] = 0$. Expanding the definition gives:

$$\sigma(A)^2 = \max\left(\left\|\mathbb{E}[AA^\top]\right\|, \left\|\mathbb{E}[A^\top A]\right\|\right).$$

First,

$$\left\|\mathbb{E}[AA^\top]\right\| = \sup_{\|v\|=1} \mathbb{E}[v^\top AA^\top v] = \sup_{\|v\|=1} \mathbb{E}\left[\left\|A^\top v\right\|^2\right] = \sup_{\|v\|=1} \sum_{i=1}^{s} \mathbb{E}\left[(e_i^\top A^\top v)^2\right] \leq s\sigma_*(A)^2.$$

Performing the same calculation for $A^\top$ in place of $A$ gives that $\left\|\mathbb{E}[AA^\top]\right\| \leq d\sigma_*(A)^2$. Combining these inequalities gives the desired result. ∎

**Corollary 3.** *Let $Y = \frac{1}{n}\sum_{i=1}^{N} Z_i$ where $Z_i \in \mathbb{R}^{d \times d}$ are mean zero independent random matrices. Assume that for some $R, k$, $\|Z_i\|_{op} \leq Rt^{k/2}$ with probability at least $1 - e^{-t}$ for all $t \geq 0$. Then if $n = d^{1+\epsilon}$ with $\epsilon > 0$ and $c = \min(1, \frac{\epsilon}{k+4})$, then for all $p \leq d^c$,*

$$\mathbb{E}\left[\|Y\|_{op}^p\right]^{1/p} \leq C\max\left(\sigma_*(Z), \frac{R}{d}\right)\sqrt{\frac{d}{n}}.$$

*where $C$ is an absolute constant.*

*Proof.* First by a union bound, we have that $\max_i \|Z_i\|_2 \lesssim Rt^k$ with probability at least $1 - ne^{-t}$. Substituting this and Lemma 13 into Lemma 12 gives that with probability at least $1 - 2ne^{-t}$,

$$\|Y\| \leq C\max\left(\sigma_*(Z), \frac{R}{d}\right)\left[\sqrt{\frac{d+t}{n}} + \left(\frac{dt^{k/2}}{n}\right)^{1/3}\left(\frac{d}{n}\right)^{1/3}t^{2/3} + \frac{dt^{\frac{k}{2}+1}}{n}\right].$$

We can factorize this by pulling out the $\sqrt{d/n}$ and using $n \geq d^{1+\epsilon}$:

$$\|Y\| \leq C\max\left(\sigma_*(Z), \frac{R}{d}\right)\sqrt{\frac{d}{n}}\left[1 + \frac{t^{1/2}}{d^{1/2}} + \frac{t^{\frac{k+4}{6}}}{d^{\frac{\epsilon}{6}}} + \frac{t^{\frac{k}{2}+1}}{n^{\frac{\epsilon}{2}}}\right].$$

We can convert this to an $p$-norm bound for $p \geq \log n$

$$\mathbb{E}[\|Y\|^p]^{1/p} \leq C\max\left(\sigma_*(Z), \frac{R}{d}\right)\sqrt{\frac{d}{n}}\left[1 + \frac{p^{1/2}}{d^{1/2}} + \frac{p^{\frac{k+4}{6}}}{d^{\frac{\epsilon}{6}}} + \frac{p^{\frac{k}{2}+1}}{n^{\frac{\epsilon}{2}}}\right].$$

Now if $p = d^c$ where $c = \min(1, \frac{\epsilon}{k+4})$ then the error terms are all less than 1 so we are done. ∎

**Lemma 14.** *For any $p \geq 2$, $\left\|\|\zeta_k(Y)\|_F\right\|_p^2 \leq (p-1)^k\binom{r+k-1}{k} \leq ((p-1)r)^k$.*

*Proof.* By Jensen's inequality and Gaussian hypercontractivity we have

$$\left\|\|\mathbb{E}[He_k(Z)|Y]\|_F\right\|_p^2 \leq \mathbb{E}[\|He_k(Z)\|_F^p]^{1/p}$$
$$\leq (p-1)^k\,\mathbb{E}\|He_k(Z)\|_F^2$$
$$= (p-1)^k k!\binom{r+k-1}{k}.$$

Dividing by $k!$ to revert to the normalized Hermite polynomials $\{h_k\}$ completes the proof. ∎

We will now bound the low-degree density ratio between the joint distribution of $(X, Y)$ and the null distribution $\mathbb{P}_0 := \mathbb{P}_X \otimes \mathbb{P}_Y$:

**Lemma 15.** *Let $\mathbb{P}_0 := \mathbb{P}_X \otimes \mathbb{P}_Y$ be the null distribution and let $\mathcal{P}_{\leq D}$ denote the orthogonal projection onto polynomials in $X$ of degree at most $D$. Then:*

$$\mathcal{P}_{\leq D}\left(\frac{d\mathbb{P}}{d\mathbb{P}_0}\right)[X, Y] = \sum_{k=0}^{D} \langle h_k(Z), \zeta_k(Y)\rangle$$

*and*

$$\left\|\mathcal{P}_{\leq D}\left(\frac{d\mathbb{P}}{d\mathbb{P}_0}\right)\right\|_2^2 \leq \binom{r+D}{r}.$$

*Proof.* Note that the density ratio is invariant to $X$ conditioned on $Z$ so we can Hermite expand directly in $Z$:

$$\mathbb{E}_0\left[h_k(Z)\frac{d\mathbb{P}}{d\mathbb{P}_0}[X,Y]\Big|Y\right] = \mathbb{E}\left[h_k(Z)\Big|Y\right] = \zeta_k(Y)$$

which implies that the Hermite coefficients of $\frac{d\mathbb{P}}{d\mathbb{P}_0}$ in $Z$ are given by $\zeta_k$. For the second equality, we have by Lemma 14:

$$\left\|\mathcal{P}_{\leq D}\left(\frac{d\mathbb{P}}{d\mathbb{P}_0}\right)\right\|_2^2 = \sum_{k=0}^{D}\mathbb{E}\|\zeta_k(Y)\|_F^2 \leq \sum_{k=0}^{D}\binom{r+k-1}{k} = \binom{r+D}{D}.$$

∎

# E   Proofs of Section 5

*Proof of Proposition 4.* We write $y = \sigma(z)$ to denote the deterministic link functions above.

1. Let $S = \{z \in \mathbb{R}^r; \sigma(z) = +1\}$. Let $k < r$ and consider $\mathbb{E}[\mathbf{h}_k(z)|z \in S] = 2\,\mathbb{E}[\mathbf{h}_k(z)\mathbf{1}(z \in S)]$. Any coordinate of this tensor corresponds to a multivariate Hermite polynomial $h_{\beta_1}(z_1)\dots h_{\beta_r}(z_r)$, with $\beta_1 + \cdots + \beta_r = k$. Since $k < r$, there must exist a coordinate $j$ s.t. $\beta_j = 0$. By noting that $\mathbb{E}_{z_j}\mathbf{1}(z \in S) \equiv 1$ and $\mathbb{E}_{z_{-j}}[h_{\beta_1}(z_1)\dots h_{\beta_r}(z_r)] = 0$, we conclude that $\mathbb{E}[\mathbf{h}_k(z)\mathbf{1}(z \in S)] = 0$ whenever $k < r$, and analogously for $\mathbb{R}^r \setminus S$. Finally, we easily verify that $\mathbb{E}[\sigma(z)z_1z_2\dots z_r] > 0$, which shows that $l^\star = r$ and hence $k^\star = r$.

2. This follows directly from $k^\star \leq l^\star$.

3. Define $K$ as the intersection of the half-spaces, determined by normals $v_1, \dots, v_M$. From the assumption that P is a $r$-dimensional multi-index model, $V = [v_1 \dots v_M]$ has rank $r$. Any unit norm vector $u \in \text{span}(V)$ thus satisfies $\max_i |v_i \cdot u| \geq \epsilon > 0$ for some $\epsilon > 0$. Let $\Sigma_K = \mathbb{E}[zz^\top|z \in K] - \mathbb{E}[z|z \in K]\mathbb{E}[z|z \in K]^\top$ be the covariance conditional on $K$. From [Klivans et al., 2024, Lemma B1], [Vempala, 2010, Lemma 4.7], for any $u$ as above it holds that $u^\top\Sigma_K u < 1$, which implies that $\text{span}(\Lambda_1) \oplus \text{span}(\Lambda_2) = \mathbb{R}^r$.

4. The argument appears already in Chen and Meka [2020], but we reproduce it here in our language for completeness.

   We will use induction over the leaps. Suppose first $S = \emptyset$, and consider the level sets $B_\lambda = \{z; |y| \geq \lambda\}$. Since $y = \sigma(z)$ is continuous and $\lim_{r\to\infty}|\sigma(rz)| = \infty$ for any $z$, for any $R > 0$ there exists $\lambda$ such that $B_\lambda$ does not contain the ball centered at $0$ of radius $R$. Thus
   $$\text{Tr}(\mathbb{E}[ZZ^\top|Z \in B_\lambda]) = \mathbb{E}[\|Z\|^2|Z \in B_\lambda] \geq R^2\,,$$
   so if $R^2 > r$ we must have $\mathbb{E}[h_2(Z)|Z \in B_\lambda] \neq 0$, and hence $\Lambda_2 \neq 0$.

   Let us now iterate over leaps. Let $S$ be the span of $\Lambda_2$. We now consider the sets
   $$B_{\lambda,\eta,S} = \{\bar{z}_S; |y| \geq \lambda, \|z_S\| \leq \eta\} \subset S^\perp\,.$$

   By now viewing $\sigma(z) = \sigma(\bar{z}_S, z_S)$ as a polynomial in $\bar{z}_S$, we again argue that for any $R > 0$ there exists $\lambda$ such that $B_{\lambda,\eta,S}$ does not contain a ball of radius $R$, and therefore we can identify another direction using the previous argument. Iterating this procedure until $S$ spans the whole $\mathbb{R}^r$ shows that $k^\star \leq 2$.

∎

*Proof of Proposition 5.* Suppose towards contradiction that $k^\star > 2$. Then for any $g \in L^2_{\mathsf{P}_y}$ we have $\mathbb{E}[g(\sigma(z))H_2(z)] = 0$. Applying the coarea formula we obtain

$$0 = \int g(y)\left(\int_{\sigma^{-1}(y)}\frac{\mathbf{h}_2(z)\gamma(z)}{\|\nabla\sigma(z)\|}d\mathcal{H}^{r-1}(z)\right)dy\,, \tag{20}$$

where $\mathcal{H}^k$ is the $k$-dimensional Hausdorff measure. Since this must be true for any measurable $g$, we conclude that

$$L(y) := \int_{\sigma^{-1}(y)} \frac{\mathbf{h}_2(z)\gamma(z)}{\|\nabla\sigma(z)\|} d\mathcal{H}^{r-1}(z) = 0 \qquad \mathsf{P}_y - \text{a.e.} \tag{21}$$

We write $\sigma(z) = \sum_{R\in\mathcal{R}}(v_R^\top z + b_R) \cdot \mathbf{1}(z \in R)$, where $\mathcal{R}$ are the different linear regions.

**Case $r = 1$:** Suppose first that there exists $\bar{y}$ and $\epsilon > 0$ such that the level sets $\sigma^{-1}(u)$ contain no critical points for $u \in (\bar{y} - \epsilon, \bar{y} + \epsilon)$. The level sets $\sigma^{-1}(u)$ are discrete, and we claim that we can represent them as

$$\sigma^{-1}(u) = \{t_i + \theta_i(u - \bar{y})\} \,, \text{ where } \sigma^{-1}(\bar{y}) = \{t_i\}_{i\in\mathcal{I}} \,,$$

and $\theta_i = 1/\sigma'(t_i) \neq 0$ have alternating sign. We thus have, for $u \in (\bar{y} - \epsilon, \bar{y} + \epsilon)$,

$$L(u) = \sum_{i\in\mathcal{I}} |\theta_i| h_2(t_i + \theta_i(u - \bar{y}))\gamma(t_i + \theta_i(u - \bar{y})) \,. \tag{22}$$

Let us integrate this quantity twice now. Using the fact that $(h_{k-1}\gamma)' = -h_k\gamma$, we have

$$\bar{L}(u) := \int_{\bar{y}-\epsilon}^u L(v)dv \tag{23}$$

$$= -\sum_{i\in\mathcal{I}} \text{sign}(\theta_i) h_1(t_i + \theta_i(u - \bar{y}))\gamma(t_i + \theta_i(u - \bar{y})) + C \,, \tag{24}$$

$$\tilde{L}(u) := \int_{\bar{y}-\epsilon}^u \bar{L}(v)dv \tag{25}$$

$$= \sum_{i\in\mathcal{I}} \text{sign}(\theta_i)\theta_i^{-1}\gamma(t_i + \theta_i(u - \bar{y})) + C(u - \bar{y} + \epsilon) + \tilde{C} \tag{26}$$

$$= \sum_{i\in\mathcal{I}} |\theta_i|^{-1}\gamma(t_i + \theta_i(u - \bar{y})) + C(u - \bar{y} + \epsilon) + \tilde{C} \,. \tag{27}$$

From $L(u) = 0$ a.e. on $(\bar{y} \pm \epsilon)$ we have $\tilde{L}(u) = 0$ for all $u \in (\bar{y} \pm \epsilon)$, leading to

$$\sum_{i\in\mathcal{I}} |\theta_i|^{-1}\gamma(t_i + \theta_i(u - \bar{y})) = -C(u - \bar{y} + \epsilon) - \tilde{C} \,, \forall u \in (\bar{y} \pm \epsilon) \,. \tag{28}$$

Since all terms are analytic, we must have this equality for all $u$, which implies $C = \tilde{C} = 0$, but this is a contradiction, since the LHS is a sum of positive terms.

**Case $r > 1$** We can represent a piece-wise linear continuous function in terms of a simplex triangulation, and the values of the function at its vertices. Consider $M = \sup_z\{\sigma(z)\}$ the maximum of $\sigma$, attained at a discrete set of global maxima. Now, let us start decreasing the level set until we reach another vertex, to say $M'$. We will study the family of level sets $\sigma^{-1}(y)$ for $y \in [M', M]$. We reparametrize $y$ as $y = M + u(M' - M)$, so this family can now be indexed with $u \in [0, 1]$.

For $\theta \in \mathbb{S}^{r-1}$ and $t \in \mathbb{R}$, let $E(\theta, t) := \{z; \theta^\top z = t\}$ denote a hyperplane normal to $\theta$ and passing at distance $t$ to the origin. We can then write

$$\sigma^{-1}(u) = \oplus_{R\in\bar{\mathcal{R}}} S_R(u) \,,$$

where $S_R(u) = E(v_R/\|v_R\|, \|v_R\|u + b_R) \cap R$, and where $\bar{\mathcal{R}}$ is the subset of linear regions crossed by these level sets. Note that by construction this family $\bar{\mathcal{R}}$ does not depend on $u$.

For $u \in (\epsilon, 1 - \epsilon)$, and for each $R \in \bar{\mathcal{R}}$, we have the following homotecy representation of the level set regions:

$$S_R(u) = \{\tilde{z} = x_R + u(z - x_R); z \in S_R(1)\} \,. \tag{29}$$

Here, $x_R$ denotes a local maximum of $\sigma$, which is also a vertex of the corresponding simplex region $R$. Consider now $\mathcal{L}(u) := \text{Tr}\{L(u)\}$. By introducing the local change of variables $\tilde{z} = \Psi_{R,u}(z) :=$

$x_R + u(z - x_R)$ for each region, we have

$$\mathcal{L}(u) = \sum_{R \in \bar{\mathcal{R}}} \theta_R \int_{S_R(u)} (\|z\|^2 - r)\gamma(z)dz \tag{30}$$

$$= u^r \sum_R \theta_R \int_{S_R(1)} \left( \|x_R + u(z - x_R)\|^2 - r \right) \gamma(x_R + u(z - x_R))dz , \tag{31}$$

where $\theta_R = \|\nabla\sigma(z)\|^{-1}$ for $z \in R$. Observe that $\mathcal{L}$ is analytic in $\mathbb{R}$, since it is a linear combination of products of analytic functions. By assumption we have that $\mathcal{L}(u) = 0$ for $u \in (0, 1)$, which implies that $\mathcal{L}$ should vanish everywhere. But for $u$ sufficiently large, observe that $\mathcal{L}(u)$ is a sum of strictly positive terms, which is a contradiction.

This shows that $\Lambda_2 \neq 0$. Let $S = \text{span}(\Lambda_2)$, and write $z = (z_S, \bar{z})$, $\bar{y} = (z_S, y)$. We can now again suppose towards contradiction that for any $g \in L^2(\mathsf{P}_{\bar{y}})$ we have $\mathbb{E}[g(\sigma(z), z_S)H_2(\bar{z})] = 0$. Defining $\bar{\sigma}^{-1}(\bar{y}) := \{\bar{z}; \sigma(z_S, \bar{z}) = y\}$, applying again the coarea formula leads to

$$L_S(\bar{y}) = \int_{\bar{\sigma}^{-1}(\bar{y})} \frac{H_2(\bar{z})\gamma(\bar{z})}{\|\nabla\bar{\sigma}(\bar{z})\|} d\mathcal{H}^{r-|S|-1}(z) = 0 \quad \mathsf{P}_{\bar{y}} - \text{a.e.} \tag{32}$$

The piece-wise linear, continuous structure is still preserved in $\bar{\sigma}$, and therefore by iteratively applying the previous argument shows that all directions will be captured with generative leaps of at most $k^\star \leq 2$.

$$\blacksquare$$

*Proof of Proposition 7.* Let $\sigma(z) = \sum_j a_j \rho(z_j)$. By definition, we have that (i) for any $\mathcal{T} : \mathbb{R} \to \mathbb{R}$ and any polynomial $q$ of degree $< k^\star(\rho)$, $\mathbb{E}[\mathcal{T}(\rho(z_j))q(z_j)] = 0$, and (ii) there exist a transformation $\zeta(y)$ such that $\mathbb{E}[\zeta(\rho(z_j))h_{k^\star(\rho)}(z_j)] \neq 0$.

Let us first show that $k^\star \geq k^\star(\rho)$. Suppose towards contradiction that we had a measurable $\mathcal{U}$ and multi-indices $(\beta_1, \ldots, \beta_r)$ with $|\beta| < k^\star(\rho)$ such that $\mathbb{E}[\mathcal{U}(\sigma(z))H_\beta(z)] \neq 0$. Then, denoting $H_\beta(z) = H_{\beta_{-j}}(z_{-j})h_{\beta_j}(z_j)$, we have

$$\mathbb{E}_{z_j} \left[ \mathbb{E}_{z_{-j}} \left( \mathcal{U}(\sigma(z))H_{\beta_{-j}}(z_{-j}) \right) h_{\beta_j}(z_j) \right] \neq 0 \tag{33}$$

$$\mathbb{E}_{z_j} \left[ \mathbb{E}_{z_{-j}} \left\{ \mathcal{U} \left( a_j y_j + \sum_{j' \neq j} a_{j'}\rho(z_{j'}) \right) H_{\beta_{-j}}(z_{-j}) \right\} h_{\beta_j}(z_j) \right] \neq 0 \tag{34}$$

$$\mathbb{E}_{z_j} \left[ \tilde{\mathcal{T}}_j(y_j)h_{\beta_j}(z_j) \right] \neq 0 , \tag{35}$$

where we defined the label transformation $\tilde{\mathcal{T}}_j(y) := \mathbb{E}_z \left[ \mathcal{U}(a_j y + \sum_{j' \neq j} a_{j'}\rho(z_{j'})) \right]$. We have thus reached a contradiction. Observe that the same argument also applies if one replaces $\sigma(z)$ by $\bar{\sigma}(z) = F(\rho(z_1), \ldots, \rho(z_r))$ for arbitrary $F$.

Let us now show $k^\star \leq k^\star(\rho) := k^\star_\rho$. We focus on a Hermite moment along a single variable, say $z_1$, given by $h_{k^\star_\rho}(z_1)$. Let $y = \rho(z_1)$ and $\eta = a_1^{-1}\sum_{j>1} a_j\rho(z_j)$, so $y$ and $\eta$ are independent. We will find a measurable function $\mathcal{U}$ such that

$$\mathbb{E}_{z_1, \eta} \left[ \mathcal{U}(\rho(z_1) + \eta)h_{k^\star_\rho}(z_1) \right] \neq 0. \tag{36}$$

We will consider a Fourier atom for $\mathcal{U}$ of the form $\mathcal{U}(t) = e^{i\tilde{\xi}t}$ for an appropriately chosen frequency $\tilde{\xi}$. For that purpose, let us first reproduce an argument from [Damian et al., 2024, Theorem 5.2]. By [Damian et al., 2024, Lemma F.2] there exists $g : \mathbb{R} \to [-1, 1]$ such that $\mathbb{E}[g(Y)h_{k^\star_\rho}(Z)] \neq 0$. We consider $g_R(y) := g(y)\mathbf{1}_{|y| \leq R}$. For $R$ sufficiently large, we claim that $\mathbb{E}[g_R(Y)h_{k^\star}(Z)] \neq 0$. We have that

$$|\mathbb{E}[g_R(Y)h_{k^\star}(Z)] - \mathbb{E}[g(Y)h_{k^\star}(Z)]| = \left|\mathbb{E}[g(Y)h_{k^\star}(Z)\mathbf{1}_{|y| \geq R}]\right|$$

$$\leq \sqrt{\mathbb{E}[g(Y)^2 h_{k^\star}(Z)^2]\mathbb{P}[|Y| \geq R]}$$

$$\leq \sqrt{\mathbb{E}[Y^2]/R^2} \tag{37}$$

which vanishes as $R \to \infty$. Therefore for sufficiently large $R$ we have $\mathbb{E}[g_R(Y)h_{k^\star}(Z)] \neq 0$. Now $g_R \in L^1(\mathbb{R}) \cap L^2(\mathbb{R})$. Let us consider its Fourier representation $g_R(y) = \int \hat{g}_R(\xi)e^{i\xi y}d\xi$. Then

$$\mathbb{E}[g_R(Y)h_{k^\star}(Z)] = \int \hat{g}_R(\xi)\, \mathbb{E}[e^{i\xi Y}h_{k^\star}(Z)]d\xi \,, \tag{38}$$

which shows that there must exist $\xi_0$ such that $\mathbb{E}_\mathsf{P}[e^{i\xi_0 Y}h_{k^\star}(Z)] \neq 0$. Moreover, observe that $\xi \mapsto \mathbb{E}_\mathsf{P}[e^{i\xi Y}h_{k^\star}(Z)] := \psi(\xi)$ is $\|\rho\|^2$-Lipschitz, since

$$|\psi'(\xi)| = |\mathbb{E}[iYe^{i\xi Y}h_{k^\star}(Z)]| \leq \sqrt{\mathbb{E}[Y^2]\,\mathbb{E}[\zeta_k(Y)^2]} \leq \mathbb{E}[Y^2] = \|\rho\|^2 \,, \tag{39}$$

so we can define $\epsilon > 0$ and $\delta > 0$ such that $|\psi(\xi)| \geq \delta$ for all $\xi \in (\xi_0 - \epsilon, \xi_0 + \epsilon)$.

Now, let us evaluate (36) with the Fourier atom. We have

$$\mathbb{E}_{z_1,\eta}\left[\mathcal{U}(\rho(z_1) + \eta)h_{k^\star_\rho}(z_1)\right] = \mathbb{E}_{z_1,\eta}\left[e^{i\tilde{\xi}(\rho(z_1)+\eta)}h_{k^\star_\rho}(z_1)\right] \tag{40}$$

$$= \mathbb{E}_{z_1}\left[e^{i\tilde{\xi}\rho(z_1)}h_{k^\star_\rho}(z_1)\right]\mathbb{E}_\eta[e^{i\tilde{\xi}\eta}] \tag{41}$$

$$= \psi(\tilde{\xi}) \cdot \varphi_\eta(\tilde{\xi}) \,, \tag{42}$$

where $\varphi_\eta(\xi) = \mathbb{E}_\eta[e^{i\eta\xi}]$ is the characteristic function of $\eta$. Assume, towards contradiction, that $\varphi_\eta(\xi) = 0$ for all $\xi \in (\xi_0 - \epsilon, \xi_0 + \epsilon)$. By definition, we have that $\varphi_\eta(\xi) = \prod_{j=2}^r \frac{a_1}{a_j}\varphi(\frac{a_1}{a_j}\xi)$, where $\varphi(\xi) = \mathbb{E}[e^{i\xi\rho(z)}]$ is the characteristic function of $\rho(z)$. We thus deduce that $\varphi(\xi)$ must vanish on an interval $\xi \in I \subseteq (\xi_0 - \epsilon, \xi_0 + \epsilon)$. Since all the moments $\mathbb{E}[y^k]$ exist by assumption, this means that

$$\forall\, \xi \in I,\, m \in \mathbb{N}\,,\, 0 = \varphi^{(m)}(\xi) = \mathbb{E}_{\mathsf{P}_y}[(iy)^m e^{iy\xi}] \,. \tag{43}$$

In particular, given any $f \in L^2(\mathsf{P}_y)$, with expansion $f(y) = \sum_k \alpha_k q_k(y)$, where $\{q_k\}_k$ is an orthonormal basis of polynomials, we deduce from (43) that $\mathbb{E}[f(y)e^{i\xi y}] = 0$ for any $f$, which would mean that $e^{i\xi y} = 0$ in $L^2(\mathsf{P}_y)$, which is a contradiction. We have thus shown that there must exist $\tilde{\xi} \in (\xi_0 - \epsilon, \xi_0 + \epsilon)$ where both $\varphi_\eta(\tilde{\xi})$ and $\psi(\tilde{\xi})$ are non-zero, proving (36).

This shows that $[\Lambda_{k^\star}]_{\beta_1} \neq 0$. Applying the same reasoning to $z_j, j \in [r]$ thus shows that $[\Lambda_{k^\star}]_{\beta_j} \neq 0$. On the other hand, if we consider $\beta$ with $|\beta| = k^\star$ but $\beta$ not of the form $\beta = (0, .., k^\star, 0, \ldots, 0)$, applying again (33) shows that $[\Lambda_{k^\star}]_\beta = 0$, ie $\Lambda_{k^\star}$ is diagonal. We conclude that $\mathrm{span}(\Lambda_{k^\star}) = \mathbb{R}^r$ and thus that $k^\star \leq k^\star(\rho)$.

$\blacksquare$

**Lemma 16** (Truncated and Fourier Label Transformations, [Damian et al., 2024, Theorem 5.2]). *Let* $\mathsf{P} \in \mathcal{P}(\mathbb{R} \times \mathbb{R})$ *with* $\mathsf{P}_z = \gamma$ *and let* $k^\star = k^\star(\mathsf{P})$. *Then there exists* $R_0, \xi_0, \epsilon_0 > 0$ *and* $\delta_0 > 0$, *and label transformations* $\mathcal{T}_R, \widetilde{\mathcal{T}}_\xi$ *of the form* $\mathcal{T}_R(y) = g(y)\mathbf{1}_{|y| \leq R}$ *and* $\widetilde{\mathcal{T}}_\xi(y) = e^{i\xi y}$ *such that*

$$|\mathbb{E}_\mathsf{P}[\mathcal{T}_R(Y)h_{k^\star}(Z)]| \geq \delta_0 \,, \quad and \quad \left|\mathbb{E}_\mathsf{P}\left[\widetilde{\mathcal{T}}_\xi(Y)h_{k^\star}(Z)\right]\right| \geq \delta_0$$

*for* $R \geq R_0$ *and* $|\xi - \xi_0| \leq \epsilon_0$.

As the Shallow NN becomes more overparametrised (and thus less well-conditioned), the generative leap is no longer 'related' to $k^\star(\rho)$. Indeed, provided $\rho$ is not a polynomial, we can use the shallow NN to approximate any desired link function $\tilde{\sigma}$ with prescribed generative leap exponent. Since the generative exponent is characterized as the first non-zero of an expansion, $k^\star(\tilde{\sigma}) \leq \ell$ is an *open* property, meaning it is stable to small perturbations of $\tilde{\sigma}$. This directly leads to the following:

**Proposition 9** (Generative Leap under Universal Approximation). *For any non-polynomial* $\rho$ *and any integer* $\ell \geq 1$, *there exists a shallow neural network of the form* $y = \sum_{j=1}^M a_j\rho(v_j^\top z + b_j)$ *such that* $(Z, Y) \sim \mathsf{P}$ *has generative leap exponent* $k^\star(\mathsf{P}) \leq \ell$.

*Proof of Proposition 9.* By Proposition 7, we can reduce ourselves to the univariate case. We first verify that, given $\sigma : \mathbb{R} \to \mathbb{R}$, there exists $\epsilon > 0$ such that if

$$\|\sigma - \tilde{\sigma}\|_{\gamma_r} \leq \epsilon \tag{44}$$

then $k^\star(\tilde{\sigma}) \leq k^\star(\sigma) = k^\star$.

From Lemma 16, we can use a sinusoid label transformation $\phi(y) = \cos(\xi y)$ for $\xi$ that depends on $\sigma$ such that $\mathbb{E}[\phi(\sigma(z))h_{k^\star}(z)] = C \neq 0$.

It suffices to verify that $\mathbb{E}[\phi(\tilde{\sigma}(z))h_{k^\star}(z)] \neq 0$. Let $a(z) = \sigma(z) - \tilde{\sigma}(z)$. Indeed, since $\phi$ is $\xi$-Lipschitz, we have

$$\forall z , \ \phi(\tilde{\sigma}(z)) = \phi(\sigma(z)) + \tilde{a}(z) , \tag{45}$$

with $|\tilde{a}(z)| \leq \xi|a(z)|$. Thus

$$|\mathbb{E}[\phi(\tilde{\sigma}(z))h_{k^\star}(z)] - \mathbb{E}[\phi(\sigma(z))h_{k^\star}(z)]| = |\mathbb{E}[\tilde{a}(z)h_{k^\star}(z)]| \tag{46}$$

$$\leq \|\tilde{a}\|_2 \leq \xi\|a\|_2 = \xi\|\sigma - \tilde{\sigma}\|_2 , \tag{47}$$

so if $\epsilon < C/\xi$ we have $k^\star(\tilde{\sigma}) \leq k^\star(\sigma)$.

Finally, using a standard universal approximation theorem, e.g using the integral representation

$$\sigma(z) = \int_{\mathbb{R}^3} c\rho(az + b)d\nu(a, b, c) = \mathbb{E}_{(a,b,c)\sim\nu}[c\rho(az + b)] \tag{48}$$

for $\nu \in \mathcal{P}(\mathbb{R}^3)$, we can obtain $\tilde{\sigma}$ satisfying (44) by doing a Monte-Carlo approximation.

∎

**Proposition 6** (Generative Leap under linear transformations). *Let $\sigma(z) : \mathbb{R}^r \to \mathbb{R} \in L^2(\gamma_r)$, $\sigma \neq C$, and let $\mathcal{M}_r$ denote the set of $r \times r$ real matrices.*

- *(i) For $\Theta \in \mathcal{M}_r$, define $y_\Theta = \sigma(\Theta^\top z)$. Then $(z, y_\Theta) \sim \mathsf{P}_\Theta$ satisfies $k^\star(\mathsf{P}_\Theta) \leq 2$ for every $\Theta$, except possibly for a set of $r^2$-dimensional Lebesgue measure zero,*

- *(ii) Assume that $(z, \sigma(z)) \sim \mathsf{P}$ has a single leap with generative exponent $k^\star$. Let $\Gamma : D \subseteq \mathbb{R}^s \to \mathcal{M}_r$ be any analytic map such that $I_r \in \mathrm{Im}(\Gamma)$ and $\Gamma(\theta)$ is invertible for all $\theta \in D$. For $\theta \in D$, define $y_\theta = \sigma(\Gamma(\theta)^\top z)$. Then $(z, y_\theta) \sim \mathsf{P}_\theta$ satisfies $k^\star(\mathsf{P}_\theta) \leq k^\star$ for every $\theta$, except possibly for a set of $s$-dimensional Lebesgue measure zero.*

*Proof.* We will again exploit analytic properties. Let us first prove (i).

Let us consider a threshold function of the form $T(y) = \mathbf{1}(\alpha_1 \leq y \leq \alpha_2)$, and its associated level set $\Omega = \{z; \alpha_1 \leq \sigma(z) \leq \alpha_2\}$. Suppose first that we can pick $\alpha_1, \alpha_2$ such that $\Omega$ is compact. Then we have $\Omega \subset B_0(R)$, the ball of radius $R$, for some $R > 0$.

For $M \in \mathbb{R}^{r \times r}$, consider the function

$$\phi(M) := \det\left[ M\left(\int_\Omega zz^\top e^{-\frac{1}{2}z^\top M^\top M z}dz\right)M^\top - \left(\int_\Omega e^{-\frac{1}{2}z^\top M^\top M z}dz\right)I \right] . \tag{49}$$

Let us now consider $\Sigma = \Theta^\top\Theta = V\Lambda^2 V^\top$, and $\tilde{Z} \sim \mathcal{N}(0, \Sigma)$. Observe that if $\Theta$ is invertible and $\phi(\Theta^{-1}) \neq 0$, then

$$\det[\Lambda_2(\mathsf{P}_\Theta)] = \det\left[\mathbb{E}[(zz^\top - I)T(y_\Theta)]\right]$$

$$= \det\left[\mathbb{E}[(zz^\top - I)\mathbf{1}(\Theta^\top z \in \Omega)]\right]$$

$$= \det\left[(\Theta^\top)^{-1}\mathbb{E}_{\tilde{Z}}[\tilde{Z}\tilde{Z}^\top\mathbf{1}(\tilde{Z} \in \Omega)]\Theta^{-1} - \mathbb{E}_{\tilde{Z}}[\mathbf{1}(\tilde{Z} \in \Omega)]I_r\right]$$

$$= \det[\Theta]^r \det\left[(\Theta^\top)^{-1}\left(\int_\Omega zz^\top e^{-\frac{1}{2}z^\top\Sigma^{-1}z}dz\right)\Theta^{-1} - \left(\int_\Omega e^{-\frac{1}{2}z^\top\Sigma^{-1}z}dz\right)I_r\right]$$

$$= \det[\Theta]^r\phi((\Theta^\top)^{-1}) , \tag{50}$$

showing that $\det[\Lambda_2(\mathsf{P}_\Theta)] \neq 0$, and thus $k^\star(\mathsf{P}_\Theta) \leq 2$.

Let us now argue that $\phi$ is analytic in $\mathbb{R}^{r \times r}$. Indeed, the determinant is analytic, and compositions preserve the analytic property, so it suffices to check that the functions $M \mapsto \phi_a(M) =$

$\int_\Omega e^{-\frac{1}{2}z^\top M^\top M z} dz$ and $M \mapsto \phi_b(M) = \int_\Omega z_i z_j e^{-\frac{1}{2}z^\top M^\top M z} dz$ are analytic. Since $\Omega$ is compact, we have that

$$\left| \frac{\partial^\beta \phi_a}{\partial M^\beta}(M) \right| \lesssim R^{2|\beta|} R^r , \tag{51}$$

for any multi-index $\beta$, and analogously for the terms $\phi_b$.

Now, observe that $\phi(0) = (\int_\Omega dz)^r \neq 0$, which implies that $\phi$ cannot be identically zero. From Mityagin [2015] we then deduce that $\phi$ can only vanish on a set of measure 0.

Let us now extend this argument to the setting where $\Omega$ is not compact. For any $\epsilon > 0$, we claim that $\phi(M)$ is analytic in $S = \{M \in \mathbb{R}^{r \times r}; \lambda_{\min}(M) \geq \epsilon\}$. Indeed, now $\phi$ is infinitely differentiable in $S$, and its components $\phi_a, \phi_b$ satisfy

$$\left| \frac{\partial^\beta \phi_a}{\partial M^\beta}(M) \right| \lesssim |\beta|!! \epsilon^{-2|\beta|-r} , \tag{52}$$

and similarly for $\phi_b$. Since now we have $\phi(0) = \infty$, for $\epsilon$ small enough we must have $\phi(M) \neq 0$ for some $M \in S$, which implies again that $\phi$ cannot be identically zero. It can therefore only vanish on a set of measure zero inside $S$, for any $\epsilon > 0$.

Let us now prove part (ii) by adapting the previous argument. Since here we are assuming a single leap, there exists a tensor-valued label transformation $T \in L^2(\mathbb{R}, (\mathbb{R}^r)^{\otimes k^\star}; \mathsf{P}_y)$ such that $\det[F^\top F] \neq 0$, where

$$F = \mathrm{Mat}_{r,(r)^{k^\star-1}} \left[ \mathbb{E}[T(\sigma(Z))\mathbf{h}_{k^\star}(Z)] \right] . \tag{53}$$

For $\Theta \in \mathbb{R}^{r \times r}$ let us now define

$$\psi(\Theta) := \det\left[ F(\Theta)F(\Theta)^\top \right] , \text{ with} \tag{54}$$

$$F(\Theta) = \mathrm{Mat}_{r,(r)^{k^\star-1}} \left[ \mathbb{E}[T(\sigma(\Theta^\top Z))\mathbf{h}_{k^\star}(Z)] \right] . \tag{55}$$

Using again $\Sigma = \Theta^\top \Theta = V\Lambda^2 V^\top$, and $\tilde{Z} \sim \mathcal{N}(0, \Sigma)$, we can rewrite this last expectation as

$$F(\Theta) = \mathrm{Mat}_{r,(r)^{k^\star-1}} \left[ \mathbb{E}_{\tilde{Z}}[T(\sigma(\tilde{Z}))\mathbf{h}_{k^\star}(\Theta^{-1}\tilde{Z})] \right] \tag{56}$$

$$= C|\Sigma|^{-1/2} \mathrm{Mat}_{r,(r)^{k^\star-1}} \left[ \int T(\sigma(z))\mathbf{h}_{k^\star}(\Theta^{-1}z) e^{-\frac{1}{2}z^\top (\Theta^\top \Theta)^{-1} z} dz \right] . \tag{57}$$

We argue again that $F(\Theta)$ is analytic in the domain $\{\Theta; \Sigma \succeq \epsilon I_r\}$ for any $\epsilon > 0$, since it is a linear combination of products of analytic functions. This implies that $\psi(\Theta)$ is also analytic in this domain. Finally, since the parametrization $\theta \mapsto \Gamma(\theta)$ is analytic by assumption, we deduce that $\bar{\psi} = \psi \circ \Gamma$ is also analytic. We know that $\bar{\psi}$ is not identically zero, therefore we conclude that it can only vanish on a set of $s$-dimensional Lebesgue measure zero. $\blacksquare$

