# OpenReview forum: "The Generative Leap: Tight Sample Complexity for Efficiently Learning Gaussian Multi-Index Models"
_NeurIPS.cc/2025/Conference — NeurIPS 2025 spotlight_

### Official Review · Reviewer_DPCL · 2025-06-27

**Clarity:** 3
**Significance:** 3
**Originality:** 3
**Rating:** 5
**Confidence:** 2

**Summary:**

This paper introduces the generative leap exponent, extending the generative exponent from single-index to multi-index Gaussian models. The authors establish a computational lower bound of under the Low-Degree Polynomial (LDP) framework for agnostically learning the hidden subspace. They also propose a sequential spectral estimator which achieves a matching upper bound.

**Questions:**

1. L115: k-th Hermite tensor uses bold type here but not in the following texts. The authors can also emphaize that $\gamma_d$ represents Gaussian distribution.

2. L150: What does $span(\Lambda)$ mean when $\Lambda$ is a tensor?

3. It might be good to explain the meaning of generative leap exponent, as in L175-177 for generative exponent to better understand their difference. The authors can also explain the leap decomposition (L149) in more details to understand the intuition and difficulty to generalize the leap exponent from single index models to multi-index models.

4. In Proposition 4, why do we only have the upper bound of $k^*$ but not its value?

5. The authors mention that GD needs $\Theta(d^k)$ samples. Is it suboptimal?

**Ethical Concerns:**

["NO or VERY MINOR ethics concerns only"]

**Final Justification:**

This is a strong paper and I recommend acceptance.

**Limitations:**

yes

**Quality:**

3

**Strengths And Weaknesses:**

The paper is theoretically sound and well organized. It is significant to find the tight sample complexity for Gaussian multi-index models and the authors propose a novel algorithm to match the sample complexity. Some sections could be more accessible, especially the leap decomposition which is quite dense, but in general core claims are well-supported.

---

> ### Author Rebuttal · Authors · 2025-07-30
>
> We sincerely thank the reviewer for their thoughtful feedback and address each point below.
>
> > L115: k-th Hermite tensor uses bold type here but not in the following texts. The authors can also emphaize that $\gamma_d$ represents Gaussian distribution.
>
> Thank you for catching these – we will correct them for the camera ready.
>
> > L150: What does $span(\Lambda)$ mean when $\Lambda$ is a tensor?
>
> We apologize for the confusion and have added a section to the paper clarifying our tensor notation, including the definition of $\mathrm{span}(\Gamma)$. In general we can define the span of a tensor at an index $l \in [0,2k]$ by:
> $$
> \mathrm{span}\_l(\Gamma) = \left\\{\Gamma[v\_1,\ldots,v\_{l-1},\cdot,v\_{l+1},\ldots,v\_{2k}] | v\_{1,\ldots,2k} \in S^\perp\right\\}
> $$
> This is equivalent to the span of the flattened version of $\Gamma$ where the $l$-th index is pulled out:
> $$
> \mathrm{span}\_l(\Gamma) = \mathrm{span}[M] \quad\text{where}\quad M\_{i,(j\_1,\ldots,j\_{2k-1})} = \Gamma\_{j\_1,\ldots,j\_{\ell-1},i,j\_\ell,\ldots,j\_{2k-1}}.
> $$
> Due to the symmetries in $\Lambda$, $\mathrm{span}_l(\Gamma)$ is independent of $l \in [0,2k]$ so we simply write $\mathrm{span}(\Lambda)$. In this case it's simplest to think of reshaping $\Lambda$ as an element of $S^\perp \otimes (S^\perp)^{\otimes 2k-1}$ and then computing the span of this matrix.
>
> > It might be good to explain the meaning of generative leap exponent, as in L175-177 for generative exponent to better understand their difference. The authors can also explain the leap decomposition (L149) in more details to understand the intuition and difficulty to generalize the leap exponent from single index models to multi-index models.
>
> We agree with the reviewer's recommendation and will include some explicit examples for the filtration corresponding to the generative leap and how it differs from the information leap. We believe that a good example to highlight is $y = h_5(z_1) + \mathrm{sign}(z_1 z_2)$. The filtration for the generative leap is $\emptyset \subset \mathrm{span}[z_1] \subset \mathrm{span}[z_1,z_2]$ since we can apply a label transformation to lower the exponent in the $z_1$ direction to $1$ (for example $y \leftarrow y^3$ works here), and then condition on $z_1$ to learn $z_2$. This corresponds to two leaps of size $1$. On the other hand, the filtration for the information leap is just $\emptyset \subset \mathrm{span}[z_1,z_2]$ directly since directly learning $h_5(z_1)$ is leap 4 so the learner is forced to learn both coordinates from $\mathrm{sign}(z_1z_2)$, which is a single leap of size $2$.
>
> > In Proposition 4, why do we only have the upper bound of $k^*$ but not its value?
>
> Proposition 4 uniquely determines $k^\star = k$ for Gaussian $k$-parity and $k^\star = 1$ for staircase functions. For intersections of halfspaces and general polynomials, we showed $k^\star \in \{1,2\}$ and this depends on the specific halfspaces/polynomials. For example, in the single-index setting, $k^\star = 1$ for any polynomial unless it is even, i.e. $p(x) = p(-x)$, in which case $k^\star = 2$. Thus, $p(x) = x^4 + x^2$ has $k^\star = 2$ while $p(x) = x^4 + x^2 + x$ has $k^\star = 1$.
>
> > The authors mention that GD needs $\Theta(d^k)$ samples. Is it suboptimal?
>
> We believe the reviewer is referring to our comment on line 50. Our goal here was simply to set the stage that the sample complexities in this line of work are of the form $n = d^p$ where the exponent $p$ depends on both the algorithm and the link function (e.g. $p = \max(1,\ell^\star-1)$ for online SGD and $p = \max(1,\ell^\star/2)$ for smoothed online SGD). We unfortunately re-used the letter $k$ in the manuscript for the generative exponent which may have led to confusion in line 50 so we will rename $k \to p$ for this line. We are still unsure what the optimal rate for gradient descent is and whether it can achieve $n = d^{\max(1,k^\star-1)}$ sample complexity with batch reuse for general multi-index models.

---

> > ### Comment · Reviewer_DPCL · 2025-08-02
> >
> > Thank you for your responses. I'm happy to keep my score.

---

### Official Review · Reviewer_xViU · 2025-06-29

**Clarity:** 3
**Significance:** 4
**Originality:** 4
**Rating:** 6
**Confidence:** 4

**Summary:**

This paper studies the complexity of learning Gaussian multi-index models, where the authors generalize the notion of the generative exponent for single-index models to generative leap exponent. They prove that this exponent controls the low-degree-polynomial lower bound for sample complexity, and prove a polynomial-time algorithm with the same sample complexity as a function of $d$. They thus show that the generative leap exponent provides a tight characterization of the sample complexity for efficient learning of Gaussian multi-index models in high dimensions. The authors also establish links between the generative and information leap exponents, and show that their framework covers several examples of popular Gaussian multi-index models studied in the literature.

**Questions:**

* What does $\mathrm{span}(\Gamma)$ mean for a tensor $\Gamma \in (S^\perp)^{\otimes 2k}$?

* I believe the union in Equation (1) is meant to be direct sum? In particular $S_i$ live in different subspaces, I’m not sure how one can take unions of them and end up with a new subspace.

* I’m a bit confused about Proposition 2. It seems that $\mathcal{T}$ acts on $(y,z_S)$, then shouldn't the push forward map be $\mathrm{Id}\_{\bar{z}\_S} \otimes \mathcal{T}$ instead of $\operatorname{Id}\_z \otimes \mathcal{T}$?

* From my understanding, not every subspace $S$ of the index space $U^\*$ is seen when defining the generative leap. However, for the lower bound of Section 3, are we free to choose any subspace $\bar{S}$ of $U^\*$? Then would we be able to choose a subspace whose leap exponent is even larger than the one seen during Definition 3? I don’t see why we would necessarily pick the one that corresponds to the maximizer of Definition 3.

* The authors mention in the paragraph starting from Line 25 that to achieve the information-theoretically optimal sample complexity of $d$, one needs to have a brute force estimator over an $\epsilon$-net which will necessarily have an exponential computational complexity in high dimensions. However, if we assume the Gaussian multi-index model has a structured covariance, it is possible to obtain an improved computational complexity (even sub-exponential) while maintaining the same linear sample complexity (in some lower effective dimension). [1] presents an example of how gradient-based learning can achieve this computational adaptivity for multi-index models (which is absent from fixed-grid optimization over $\epsilon$ nets). It is generally interesting to know how the picture here would change once we have a structured covariance, which is known to improve the dependence on information exponent for algorithms in the correlational statistical query class [2, 3].

* The discussion in Section 4.2 can of course be extended to non-polynomial time algorithms. Without any assumptions on $P$, learning the regression function when $Y$ depends on $r$ directions could have a minimax optimal rate that looks like $(1/n)^{O(1/r)}$.

* The authors may want to use $s_i$ in the 3rd line of Algorithm 1.

References:

[1] A. Mousavi-Hosseini et al. "Learning Multi-Index Models with Neural Networks via Mean-Field Langevin Dynamics." ICLR 2025.

[2] J. Ba et al. "Learning in the presence of low-dimensional structure: a spiked random matrix perspective." NeurIPS 2023.

[3] A. Mousavi-Hosseini et al. "Gradient-Based Feature Learning under Structured Data." NeurIPS 2023.

**Ethical Concerns:**

["NO or VERY MINOR ethics concerns only"]

**Final Justification:**

Per my original review this is a strong contribution and all reviewers are in agreement.

**Limitations:**

yes

**Paper Formatting Concerns:**

No concerns.

**Quality:**

4

**Strengths And Weaknesses:**

This is a technically strong paper with many novel and interesting results, and provides a complete picture of the complexity of learning Gaussian multi-index models, a problem that has received a lot of attention from the community as an example where learners can adapt to a latent low-dimensional structure in a high-dimensional ambient space and perform feature learning with non-linear predictors.

 The paper is generally well-written and easy to read, but there are a few points that can improve the readability. In particular, I think that the readers could have a better intuition of the constructions in Section 2 if the authors present the example of staircase functions and how it leads to concrete instantiations of Definitions 3, 4, and Proposition 1. Also the authors could explain a bit more about certain notations, discussed below.

---

> ### Author Rebuttal · Authors · 2025-07-30
>
> We sincerely thank the reviewer for their thoughtful feedback and address each point below.
>
> > The paper is generally well-written and easy to read, but there are a few points that can improve the readability. In particular, I think that the readers could have a better intuition of the constructions in Section 2 if the authors present the example of staircase functions and how it leads to concrete instantiations of Definitions 3, 4, and Proposition 1.
>
> We agree with the reviewer's recommendation and will include some explicit examples for the filtration corresponding to the generative leap and how it differs from the information leap. We believe that a good example to highlight is $y = h_5(z_1) + \mathrm{sign}(z_1 z_2)$. The filtration for the generative leap is $\emptyset \subset \mathrm{span}[z_1] \subset \mathrm{span}[z_1,z_2]$ since we can apply a label transformation to lower the exponent in the $z_1$ direction to $1$ (for example $y \leftarrow y^3$ works here), and then condition on $z_1$ to learn $z_2$. This corresponds to two leaps of size $1$. On the other hand, the filtration for the information leap is just $\emptyset \subset \mathrm{span}[z_1,z_2]$ directly since directly learning $h_5(z_1)$ is leap 4 so the learner is forced to learn both coordinates from $\mathrm{sign}(z_1z_2)$, which is a single leap of size $2$.
>
> > What does $\mathrm{span}(\Gamma)$ mean for a tensor $\Gamma \in (S^\perp)^{\otimes 2k}$?
>
> We apologize for the confusion and have added a section to the paper clarifying our tensor notation, including the definition of $\mathrm{span}(\Gamma)$. In general we can define the span of a tensor at an index $l \in [0,2k]$ by:
> $$
> \mathrm{span}\_l(\Gamma) = \left\\{\Gamma[v\_1,\ldots,v\_{l-1},\cdot,v\_{l+1},\ldots,v\_{2k}] | v\_{1,\ldots,2k} \in S^\perp\right\\}
> $$
> This is equivalent to the span of the flattened version of $\Gamma$ where the $l$-th index is pulled out:
> $$
> \mathrm{span}\_l(\Gamma) = \mathrm{span}[M] \quad\text{where}\quad M\_{i,(j\_1,\ldots,j\_{2k-1})} = \Gamma\_{j\_1,\ldots,j\_{\ell-1},i,j\_\ell,\ldots,j\_{2k-1}}.
> $$
> Due to the symmetries in $\Lambda$, $\mathrm{span}_l(\Gamma)$ is independent of $l \in [0,2k]$ so we simply write $\mathrm{span}(\Lambda)$. In this case it's simplest to think of reshaping $\Lambda$ as an element of $S^\perp \otimes (S^\perp)^{\otimes 2k-1}$ and then computing the span of this matrix.
>
> > I believe the union in Equation (1) is meant to be direct sum? In particular $S_i$ live in different subspaces, I’m not sure how one can take unions of them and end up with a new subspace.
>
> Yes, you are correct that equation (1) should be a direct sum. We used $S \cup S'$ to represent $S \oplus S'$ at various points throughout the paper and will correct this for the camera ready.
>
> > I’m a bit confused about Proposition 2. It seems that $\mathcal{T}$ acts on $(y,z_S)$, then shouldn't the push forward map be $\mathrm{Id}_{\bar{z}_S} \otimes \mathcal{T}$ instead of $\operatorname{Id}_z \otimes \mathcal{T}$?
>
> The pushforward map should indeed read $\mathrm{Id}\_{\overline{z}\_S} \otimes \mathcal{T}$, rather than $\mathrm{Id}\_{\overline{z}\_S} \otimes \mathcal{T}$. Thank you for catching this typo, we will correct it in our next revision.
>
> > From my understanding, not every subspace $S$ of the index set $U^\star$ is seen when defining the generative leap. However, for the lower bound of Section 3, are we free to choose any subspace $\bar{S}$ of $U^\star$? Then would we be able to choose a subspace whose leap exponent is even larger than the one seen during Definition 3? I don’t see why we would necessarily pick the one that corresponds to the maximizer of Definition 3.
>
> Theorem 1 is stated for picking the subspace $\overline{S}$ that corresponds to the filtration just before the leap corresponding to the generative leap (specifically, a choice of $\overline{S}$ that is seen in Definition 3). However we could easily generalize it to the following statement: for any subspace $S$, distinguishing $\mathbb{H}_0,\mathbb{H}_1$ requires $n \gtrsim d^{k(\overline{S})/2}$ samples. As $\overline{S}$ was chosen such that $k(\overline{S}) = k^\star$, this would imply Theorem 1.
>
> You are indeed correct that we could apply this to any subspace $\overline{S}$, including those not seen during Definition 3. Arguing that $n \ge d^{k^\star/2}$ samples are necessary for *estimation*, therefore follows from this inductive argument:
>
> - At the start you have no information ($\overline{S} = \emptyset$), and you can prove that you need $n \gtrsim d^{k(\emptyset)/2} = d^{k_1/2}$ samples to detect the planted model, and therefore to estimate any additional directions.
> - If the learner has more than $d^{k_1/2}$ samples, we will generously provide them with all of $S_1$. Given $S_1$, detecting the planted model learning any additional directions requires $n \gtrsim d^{k(S_1)/2} = d^{k_2/2}$ samples.
> - ...
>
> This induction shows that recovering any direction in $S_{i+1} \setminus S_i$ requires $n \gtrsim d^{\max(k_1,\ldots,k_i)/2}$ samples and since $k^\star = \max_i k_i$, it demonstrates that recovering the index set requires $n \gtrsim d^{k^\star/2}$ samples. We will add the generalized version of Theorem 1 in terms of $k(S)$ and the discussion connecting estimation with this series of decision problems to Section 3.
>
> > The authors mention in the paragraph starting from Line 25 that to achieve the information-theoretically optimal sample complexity of $d$, one needs to have a brute force estimator over an -net which will necessarily have an exponential computational complexity in high dimensions. However, if we assume the Gaussian multi-index model has a structured covariance, it is possible to obtain an improved computational complexity (even sub-exponential) while maintaining the same linear sample complexity (in some lower effective dimension). [1] presents an example of how gradient-based learning can achieve this computational adaptivity for multi-index models (which is absent from fixed-grid optimization over  nets). It is generally interesting to know how the picture here would change once we have a structured covariance, which is known to improve the dependence on information exponent for algorithms in the correlational statistical query class [2, 3].
>
> We definitely agree with the reviewer that the picture gets much more interesting when there is additional structure in the input data. The generative leap directly extends to the setting of non-isotropic covariances so long as this covariance is fixed in advance. For example, for a single index model if $x \sim N(0,\Sigma)$ and $w^\star$ is chosen such that $\mathbb{E}_x (x \cdot w^\star)^2 = 1$, then we can just transform $(x,w^\star) \to (\Sigma^{-1/2} x, \Sigma^{1/2} w^\star)$ and then $x$ will be isotropic and $w^\star$ will be uniform on the sphere so the generative exponent still applies. However, if $\Sigma$ and $w^\star$ are correlated (as in [2,3]), then the learner can certainly leverage this information to learn more efficiently. We will add a section in the paper discussing possible extensions of our framework to more structured inputs.
>
> > The discussion in Section 4.2 can of course be extended to non-polynomial time algorithms. Without any assumptions on $P$,  learning the regression function when $Y$ depends on $r$ directions could have a minimax optimal rate that looks like $1/n^{O(1/r)}$.
>
> We agree that the minimax rate for estimation is $1/n^{O(1/r)}$. However, it's not immediately clear that this is necessary for subspace recovery. One could potentially hope that subspace recovery is an easier task. The purpose of section 4.2 is to give an example where it is actually necessary to find the right function on $z_1,\ldots,z_{r-1}$ before you can apply the right label transformation to identify $z_r$.
>
> > The authors may want to use $s_i$ in the 3rd line of Algorithm 1.
>
> Could you clarify which line you are referring to? Are you referring to the line $y \gets [y,\Pi_S x] \in \mathbb{R}^{|S| + 1}$ in Algorithm 2?

---

> > ### Comment · Reviewer_xViU · 2025-08-02
> >
> > Thank you for the detailed responses. I think this is a strong contribution and I'm happy to keep my score.
> >
> > > Could you clarify which line you are referring to? Are you referring to the line $y \gets [y,\Pi_S x] \in \mathbb{R}^{|S| + 1}$ in Algorithm 2?
> >
> > I was referring to the fact that we are taking the top $s$ eigenvectors in Algorithm 1, but then I noticed this is already being pointed out in the output of the algorithm.

---

### Official Review · Reviewer_buZ9 · 2025-06-30

**Clarity:** 3
**Significance:** 4
**Originality:** 3
**Rating:** 5
**Confidence:** 4

**Summary:**

The authors investigated the computationally optimal sample complexities to learn the intrinsic features of multi-index models.

The authors first generalized the generative exponent (from single index models) to generative leap exponent $k^\star$ with leap decomposition, which relates to the nested easy-to-hard subspace structure & leap exponent.

Then they characterized the hardness of learning a multi-index models by proving that a sample complexity of $\Omega(d^{k^\star/2} )$ is essential for the low-degree polynomial test to have non-trivial power in detecting the planted subspace.

For a matching upper bound, the authors propose to use a U-statistics that resembles the unfolding technique in the multi-spike tensor PCA and proved that $O(d^{k^\star/2})$ is essential for this algorithm to learn the planted direction.

**Questions:**

1. In the upper bound algorithm, you proposed a kernel-based algorithm that implements the label transformation of potentially infinite order and the conditions for the kernel seems to be very weak. However, in the problem of learning single-index models, batch-reusing / explicit label transformation are deployed to go beyond the CSQ queries[1,2] and certain conditions are required for general gradients to attain the optimal sample complexities [e.g. Assumpion 4.1 (b), 2]. Does this kernel trick automatically select such label transformations? Or is there a way that we can use this method in iterative algorithms to achieve optimal guarantee?
2. Does the sub-space filtration (or flag, in (1)) from the generative leap exactly equal to the filtration defined for the information leap?  To put in other way, does the direction that is easier to learn in the notion of generative exponent necessarily easier to learn in the notion of information exponent?

[1] Lee, Jason D., Kazusato Oko, Taiji Suzuki, and Denny Wu. "Neural network learns low-dimensional polynomials with sgd near the information-theoretic limit." Advances in Neural Information Processing Systems 37 (2024): 58716-58756.

[2] Chen, Siyu, Beining Wu, Miao Lu, Zhuoran Yang, and Tianhao Wang. "Can neural networks achieve optimal computational-statistical tradeoff? an analysis on single-index model." In The Thirteenth International Conference on Learning Representations. 2025.

**Ethical Concerns:**

["NO or VERY MINOR ethics concerns only"]

**Limitations:**

yes

**Paper Formatting Concerns:**

No.

**Quality:**

4

**Strengths And Weaknesses:**

Pros
1. The paper is mathematically rigorous and well organized.
2. The content is quite complete and novel, by fully settling the low-degree detection lower bound and a matching upper bound. Several examples are provided to help readers with understanding the notion of generative leap and the hardness of the multi-index models.
3. The proof sketch for the upper bound is very clean, easy to follow and sufficient to cover the main idea.

Cons
1. The layout, definitions and results seem to be a bit dense for readers.

---

> ### Author Rebuttal · Authors · 2025-07-30
>
> We sincerely thank the reviewer for their thoughtful feedback and address each point below.
>
> > In the upper bound algorithm, you proposed a kernel-based algorithm that implements the label transformation of potentially infinite order and the conditions for the kernel seems to be very weak. However, in the problem of learning single-index models, batch-reusing / explicit label transformation are deployed to go beyond the CSQ queries[1,2] and certain conditions are required for general gradients to attain the optimal sample complexities [e.g. Assumpion 4.1 (b), 2]. Does this kernel trick automatically select such label transformations? Or is there a way that we can use this method in iterative algorithms to achieve optimal guarantee?
>
> Yes, this kernel trick automatically selects such label transformations. In fact, what this implicitly relies on is that it suffices to use a *random* label transformation. For example:
>
> - uniformly sample $q \in [0,1]$, and then threshold the labels $y_1,\ldots,y_n$ at the $q$-th quantile, i.e. if $y_1 \le \ldots, \le y_n$ then $y_1,\ldots,y_{\lfloor q n \rfloor} = 0$ and $y_{\lfloor qn \rfloor}, \ldots, y_n = 1$ (min CDF kernel)
>
> - transform $y \to \cos(w y + b)$ where $w \sim N(0,1)$ and $b \sim \mathrm{Unif}([0,2\pi])$ (RBF kernel)
>
> Either of these choices would succeed with probability $1$. Our kernel estimator can be interpreted as "averaging" over the randomness in such a label transformation. Therefore, to use this method in an iterative algorithm (e.g. batch re-use) it suffices to show that the dynamics apply a sufficiently random label transformation to the labels. While this has been analyzed in the single-index setting (e.g. [1,2] above), we are not aware of any results that imply learnability of general multi-index functions using gradient descent given $n \gtrsim d^{O(k^\star)}$ samples where $k^\star$ is the generative leap.
>
> > Does the sub-space filtration (or flag, in (1)) from the generative leap exactly equal to the filtration defined for the information leap? To put in other way, does the direction that is easier to learn in the notion of generative exponent necessarily easier to learn in the notion of information exponent?
>
> No, in general these filtrations can be quite different. As a simple example, consider:
> $$
> y = h_5(z_1) + \mathrm{sign}(z_1 z_2).
> $$
> The filtration for the generative leap is $\emptyset \subset \mathrm{span}[z_1] \subset \mathrm{span}[z_1,z_2]$ since we can apply a label transformation to lower the exponent in the $z_1$ direction to $1$ (for example $y \leftarrow y^3$ works here), and then condition on $z_1$ to learn $z_2$. This corresponds to two leaps of size $1$.
>
> On the other hand, the filtration for the information leap is just $\emptyset \subset \mathrm{span}[z_1,z_2]$ directly since directly learning $h_5(z_1)$ is leap 4 so the learner is forced to learn both coordinates from $\mathrm{sign}(z_1z_2)$, which is a single leap of size $2$.

---

> > ### Comment · Reviewer_buZ9 · 2025-08-02
> >
> > Thanks for your in-depth comments! This settled my questions. I will keep my scores unchanged

---

### Official Review · Reviewer_s6Px · 2025-06-30

**Clarity:** 4
**Significance:** 2
**Originality:** 2
**Rating:** 5
**Confidence:** 3

**Summary:**

The paper considers multi-index models with Gaussian inputs in high dimensions and a finite number of indices. For this model, the authors show that one can estimate the hidden subspaces in the model with a sample complexity $n = \Theta(d^{1\lor k^*/2})$, where $k^*$, the generative leap, depends on the model. This is a necessary and sufficient condition.

**Questions:**

1. In 4.2 you discussed a fine-grained analysis, in the sense of finding the minimal constant $\lim_{n,d\to\infty} n/d^{1\lor k^*/2}$ such that the hidden subspaces can be recovered. In particular you state that the dependance on the number of indices can be exponentially bad. Can you provide a more comprehensive classification?
2. In 5.3 you state that while shallow networks have a generative leap at least as large as the generative exponent, wider networks might not have this bound. Is there a practical example of this with some activations?
3. Is there some way to salvage any of this theory in the case of $r = \Theta(d)$ indices?

**Ethical Concerns:**

["NO or VERY MINOR ethics concerns only"]

**Final Justification:**

I believe this to be an incredibly solid paper. My doubts were cleared by the rebuttal and I happily raised my score to full acceptance.

**Limitations:**

Yes

**Paper Formatting Concerns:**

No concerns

**Quality:**

4

**Strengths And Weaknesses:**

The paper is detailed, clear and extremely well written. The proofs are clear, and the technique is interesting, although similar in essence to the one for generative exponents. The main contribution of this paper, the description of the generative leap, is a fundamental contribution to the field and will surely have a big impact in the theory community. As the authors correctly state, most cases of practical interest have generative exponent less then or equal to 2, which were already studied in several previous works, so its fundamental limitation lies in the limited applicability of the results.

---

> ### Author Rebuttal · Authors · 2025-07-30
>
> We sincerely thank the reviewer for their thoughtful feedback and address each point below.
>
> > In 4.2 you discussed a fine-grained analysis, in the sense of finding the minimal constant $\lim_{n,d\to\infty} n/d^{1\lor k^*/2}$ such that the hidden subspaces can be recovered. In particular you state that the dependance on the number of indices can be exponentially bad. Can you provide a more comprehensive classification?
>
> The tight constant will depend significantly on whether or not the learner has knowledge of the link function. For example, when $k^\star \le 2$, [Troiani et al. 2024] computed the tight constants for AMP, and we believe it is likely that a more careful low-degree polynomial lower bound would match these constants. Tight constants do not generally exist for $k^\star > 2$ since runtime and sample complexity can be continuously traded off. However, we can compute the exact weak-recovery threshold for our matrix U-statistic estimator when the optimal label transformation $\zeta_k$ is used. For Gaussian $k$-parity, this constant is:
> $$
> n \ge C_k d^{k/2} \quad\text{where}\quad C_k = \left(\frac{\pi}{2}\right)^k \cdot \frac{1}{\sqrt{(k-1)!}}
> $$
> which scales like $(\pi^2 e/k)^{\frac{k}{2}}$. In particular, for Gaussian $k$-parity, this constant is exponentially small in $r=k$.
>
> More generally when the learner does not have knowledge of the link function, the constants (and $r$ dependencies) will generally be worse, and the constants will depend on the prior over link functions.
>
> > In 5.3 you state that while shallow networks have a generative leap at least as large as the generative exponent, wider networks might not have this bound. Is there a practical example of this with some activations?
>
> The key difference between Proposition 6 and Proposition 7 is that the weights are restricted to be orthogonal in Proposition 6. In addition, we apologize for a typo in Proposition 7 which we have fixed in a revision – the corrected statement is that for any non-polynomial $\rho$ and integer $C \in [1,k^\star(\rho)]$, there exists a shallow neural network with $k^\star \le C$.
>
> > Is there some way to salvage any of this theory in the case of $r = \Theta(d)$ indices?
>
> Our paper is focused on the setting in which $r = O(1)$ and $d \to \infty$. When $r = \Theta(d)$, the function is in a sense "generic" since it can depend on almost all of the coordinates, and getting learning guarantees in this setting would require imposing additional assumptions on $y = f^\star(x)$. However, it may be possible to salvage the theory when $r = d^{\alpha}$ for some $\alpha < 1$. This would require developing more fine-grained estimates of the $r$-dependencies in both our upper and lower bounds. For $k^\star = 1,2$ the tight constants (assuming knowledge of the link function) were computed in [Troiani et al. 2024], however we leave extending this to larger $k^\star$ to future work.

---

> > ### Comment · Reviewer_s6Px · 2025-08-02
> >
> > Thanks for your detailed answer. I am happy to raise my rating.

---

### Decision · Program_Chairs · 2025-09-17

**Decision:**

Accept (spotlight)

**Comment:**

The goal of the paper is the analysis of learning in generic Gaussian multi-index models. The investigation generalises the notion of generative exponent to generative leap exponent, proving that this quantity is crucial in the scaling of the sample complexity scaling for learning. The paper collected unanimous positive feedbacks: its novelty and soundness have been appreciated by all Reviewers, in particular with respect to the introduction of the aforementioned generative leap exponent as informative complexity measure. As considerable effort has been put by the theoretical community in the study of single index and multi index model, the paper provides a significant step forward in such a line of research and might be a reference for further investigations on such a prototipical setups. I therefore recommend its acceptance in NeurIPS, possibly as *spotlight*.